# GPER1 links estrogens to centrosome amplification and chromosomal instability in human colon cells

Miriam Bühler, Jeanine Fahrländer, Alexander Sauter⬤, Markus Becker⬤, Elisa Wistorf⬤, Matthias Steinfath, Ailine Stolz⬤

The role of the alternate G protein–coupled estrogen receptor 1 (GPER1) in colorectal cancer (CRC) development and progression is unclear, not least because of conflicting clinical and experimental evidence for pro- and anti-tumorigenic activities. Here, we show that low concentrations of the estrogenic GPER1 ligands, 17β-estradiol, bisphenol A, and diethylstilbestrol cause the generation of lagging chromosomes in normal colon and CRC cell lines, which manifest in whole chromosomal instability and aneuploidy. Mechanistically, (xeno)estrogens triggered centrosome amplification by inducing centriole overduplication that leads to transient multipolar mitotic spindles, chromosome alignment defects, and mitotic laggards. Remarkably, we could demonstrate a significant role of estrogen-activated GPER1 in centrosome amplification and increased karyotype variability. Indeed, both gene-specific knockdown and inhibition of GPER1 effectively restored normal centrosome numbers and karyotype stability in cells exposed to 17β-estradiol, bisphenol A, or diethylstilbestrol. Thus, our results reveal a novel link between estrogen-activated GPER1 and the induction of key CRC-prone lesions, supporting a pivotal role of the alternate estrogen receptor in colon neoplastic transformation and tumor progression.

## Introduction

Colorectal cancer (CRC) is the third most common cancer worldwide and the second deadliest malignancy in both men and women. Key features of CRC are structural and numerical centrosome defects, which are an early and stable event in colon carcinogenicity and associated with poor prognosis (1, 2). Cells with amplified centrosomes transiently generate multipolar mitoses, which are prone to induce erroneous microtubule–kinetochore attachments that favor the formation of lagging anaphase chromosomes (3). It is widely accepted that lagging chromosomes represent an important mechanism for whole chromosomal instability (w-CIN) and

aneuploidy (4, 5, 6). W-CIN is referred to as the perpetual gain or loss of whole chromosomes during mitosis and may contribute to tumorigenesis, tumor progression, and therapy resistance (7, 8). Importantly, CRC represents a prime example of a tumor entity exhibiting w-CIN (9, 10) with centrosome amplification (CA) as a major underlying mechanism (1, 2).

Although several pathways underlie the etiology of CA in colorectal carcinomas (2), their upstream regulators are poorly understood. Dietary intake and environmental factors play a major role in CRC pathogenesis and may promote poor outcomes (11). Remarkably, the environmental estrogenic chemical bisphenol A (BPA) was shown to enhance the development and progression of colon cancer by modulating protein profiles related to tumorigenesis and metastasis, thereby triggering epithelial–mesenchymal transition, migration, and invasion (12, 13). Studies in rodents and cell lines derived from hormonally regulated tissues demonstrate that not only the steroidal estrogen 17β-estradiol (E2) but also synthetic endocrine active substances with estrogenic activities (i.e., xenoestrogens), including the non-steroidal estrogen diethylstilbestrol (DES) and BPA, disturb bipolar mitotic spindle formation, centrosome duplication, spindle microtubule attachment to kinetochores, and karyotype stability (14, 15, 16, 17, 18, 19, 20, 21). However, a potential link between estrogen actions and the evolution of numerical centrosome defects and w-CIN driving the pathophysiology of a non-classical hormone-regulated tissue, that is, the colon, is still missing.

Effects of (xeno)estrogens are complicated by at least three main estrogen receptors, the nuclear estrogen receptors, ERα and ERβ, and the alternate G protein–coupled seven-transmembrane estrogen receptor GPER1/GPR30 (22). ERα has either low or no expression in both normal colon and CRC cells, although splice variants do exist (23, 24, 25). ERβ seems to be the predominant nuclear estrogen receptor in the differentiated colonic epithelium, which is lost during cancer progression (26, 27). GPER1 is expressed in the gastrointestinal tract, and its activity is stimulated not only by endogenous estrogens but also by numerous xenoestrogens (e.g., bisphenols), anti-estrogens such as tamoxifen and fulvestrant (ICI182,780), pesticides (e.g., atrazine), and synthetic GPER1-selective

Department of Experimental Toxicology and ZEBET, German Federal Institute for Risk Assessment (BfR), German Centre for the Protection of Laboratory Animals (Bf3R), Berlin, Germany

Correspondence: aline.stolz@bfr.bund.de

ligands (e.g., G-1), whereas it is blocked by specific antagonists such as G15 and G36 (28). Apart from being involved in physiological processes in the colon, GPER1 also links pathophysiological aspects by regulating colonic motility, immune regulation, and inflammation in CRC-associated diseases (28). In fact, the binding of E2, BPA, and DES to GPER1 activates cancer-related pathways, which are associated with increased cell proliferation and migration, dependent on the CRC tumor microenvironment (29).

In this study, we uncover a novel (xeno)estrogen/GPER1/centrosome axis, which has an important impact on genomic stability, proposing a potential role in colon carcinogenicity. We show for the first time that the estrogenic GPER1 activators, E2, BPA, and DES, cause numerical CA triggered by centriole overduplication, leading to karyotype instability in normal colon and CRC cell lines in a GPER1-dependent manner. Given that the sex hormone E2, the well-accepted endocrine-disrupting chemical BPA (ECHA.eu, (30)), and the known carcinogen DES (31, 32, 33) trigger the evolution of key CRC-prone lesions, our results may provide important clues for a possible role of estrogens in colon pathogenesis and shed light on the underlying mechanism that involves GPER1 function.

# Results

### Estrogenic substances cause CA in colon cells

Supernumerary centrosomes represent a hallmark of CRC and are significantly involved in tumor initiation, progression, and therapy resistance (1, 2, 7, 8). However, upstream triggers are hardly known. To investigate whether (xeno)estrogens induce CA in a colon cell system, we treated CRC-derived HCT116 and CCD 841 CoN normal colon epithelial cell lines with increasing concentrations of E2, BPA, and DES. Subsequently, we determined the amount of cells with more than two centrosomal γ-tubulin signals (34) by immunofluorescence microscopy (Fig S1A and B). Because the centrosome duplication cycle follows the cell division cycle (35), cells were treated for a period of 48 h to ensure the establishment of the phenotype, which is consistent with other studies (14, 18, 21). We found a significant concentration-dependent increase in CA in both transformed and non-transformed colon cells to a saturated level at ~10 nM of E2, BPA, and DES (Fig S1A and B). Interestingly, the overall frequency of CA did not exceed ~6–7% in HCT116 and ~3–4% in CCD 841 CoN cells within a treatment period of up to 6 d (Fig S1C and D). The comparatively mild effects on CA observed in normal colon epithelial cells are consistent with a tightly regulated machinery of centrosome duplication that prevents CA in normal cells (36). The extent of CA in HCT116 cells was comparable to that of low-dose–treated prostate cancer cells (14, 21) and similar to an ectopic expression of PLK4, the master regulatory kinase of centrosome duplication ((37), Fig S1E), suggesting that basically all cells respond to the treatment. The estrogen-induced CA observed in HCT116 and CCD 841 CoN cells at low nanomolar concentrations (Fig 1A and B) was verified in additional CRC cell lines, that is, RKO and HCT-15 (Fig 1C and D), indicating a cell transformation–independent effect of (xeno)estrogens. To examine the specificity of CA because of E2, BPA, and DES, we included substances being structurally related to steroid hormones (e.g., cholesterol and dexamethasone) and the

estrogenic herbicide atrazine in our studies. Of note, none of these substances led to a significant increase in CA (Fig S1F and G). These results not only demonstrate the specificity of (xeno)estrogens to induce CA in our colon cell systems but also suggest various molecular mechanisms that might trigger CA in colon (cancer) cells upon treatment with distinct estrogenic substances.

Numerical centrosome defects are most commonly described in cancers and involve a number of mechanisms, including fragmentation of the pericentriolar material (PCM), centriole overduplication, de novo assembly of centrioles, and premature centriole disengagement, among others (35). A precise categorization of colon (cancer) cells displaying CA to three, four, or more than four centrosomes revealed mainly three PCM-containing centrosomes upon treatment with (xeno)estrogens (Fig 1A–D, representative images; and Fig 2A). To exclude PCM fragmentation after the generation of acentriolar centrosomes, which are not representative of bona fide CA (35, 38), we requantified centrosome numbers of treated cells by counting exclusively centriole-positive centrosome foci. To this end, we labeled cells with the well-characterized centrosome marker, γ-tubulin (34) as before, and included the centriole-specific Cep135 and CP110 markers (39, 40) in co-immunostainings. Importantly, we revealed that amplified centrosomes are centriole-positive upon (xeno)estrogen treatment (Figs 1E–H and S1H–K). An expected baseline of cells lacking centriole signals could be observed in HCT116 and RKO cells (41). The frequency distribution of centriole numbers based on counts with CP110 (Fig 2B) was almost identical to that with γ-tubulin (Fig 2A). The distribution of Cep135 differed from that of γ-tubulin in HCT116 and even more in CCD 841 CoN cells with respect to individual treatments (Fig 2C). These results (i) verify γ-tubulin as a suitable marker sufficient to detect supernumerary centrosomes in all further experiments, (ii) confirm bona fide CA in a non-classical hormone-regulated tissue, that is the colon, after treatment with distinct estrogenic substances, and, importantly, (iii) suggest that (xeno)estrogens might perturb the centriole duplication cycle in a colon cell system.

### (Xeno)estrogen-triggered CA involves centriole overduplication

Given that (xeno)estrogens induce centriole-positive CA excluding PCM fragmentation (Figs 1E–H and S1H–K), we sought to investigate whether E2, BPA, and DES cause centriole overduplication in the colon cell system, as recently shown in BPA-treated HeLa cells (15). First, we partially repressed a key marker for centriole duplication, that is, PLK4 (37), in (xeno)estrogen-treated HCT116 cells and examined for CA. Indeed, the partial repression of PLK4 suppressed CA in E2-, BPA-, and DES-treated cells (Fig 2D). We conclude that centriole overduplication seems to represent a promising underlying molecular mechanism for (xeno)estrogen-induced CA. However, because of its superordinate role in regulating centrosome numbers, we cannot exclude that Plk4 overlays other (xeno)estrogen-triggered mechanisms.

Centriole overduplication usually involves defects in the copy-number control leading to parental centrioles templating the assembly of more than one new centriole each (Fig 2E, (42)). At the end of mitosis, the parental centriole and its overduplicated procentrioles split apart (i.e., disengage), resulting in daughter cells with

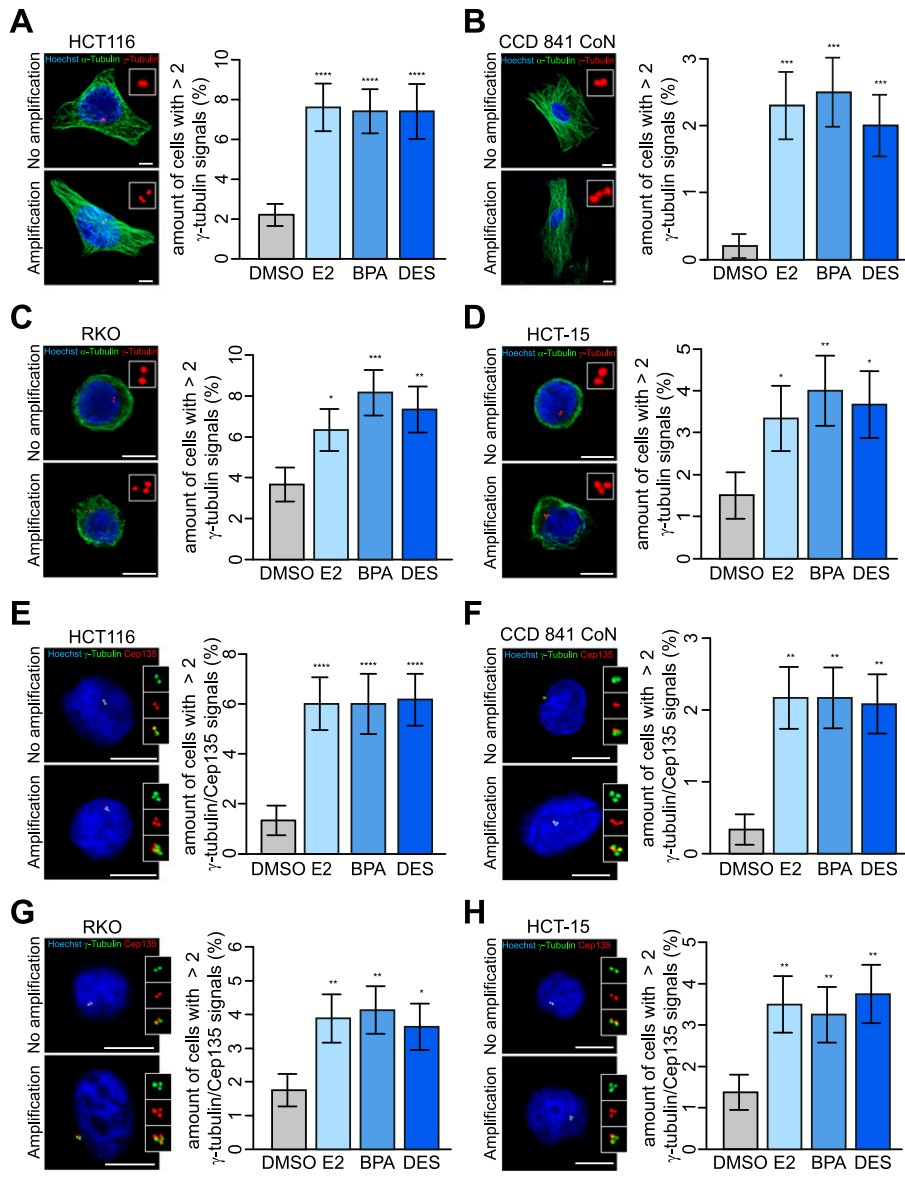

**Figure 1. 17β-Estradiol, bisphenol A, and diethylstilbestrol induce centrosome amplification in human colon cells.**

**(A, B, C, D)** Detection and quantification of interphase cells with (xeno)estrogen-triggered centrosome amplification. HCT116 (A), CCD 841 CoN (B), RKO (C), and HCT-15 cells (D) were cultured in a stripped FCS medium and treated with DMSO, or 10 nM 17β-estradiol, bisphenol A, or diethylstilbestrol for 48 h. Centrosome amplification was detected by immunofluorescence microscopy. Representative images of cells with or without amplified centrosomes are shown (centrosomes, γ-tubulin, red; microtubules, α-tubulin, green; nuclei, Hoechst 33342, blue; scale bar, 10 μm). Insets show enlarged γ-tubulin signals. The graphs show the quantification of the amount of cells with more than two γ-tubulin signals at centrosomes (mean ± s.d., n = 5 with a total of 1,000 cells (A, B) and n = 3 with a total of 600 cells (C, D)). Wald's z-statistics computed by the R function *glmmTMB* was used to calculate the *P*-value. \**P* < 0.05; \*\**P* < 0.01; \*\*\**P* < 0.001; and \*\*\*\**P* < 0.0001. **(E, F, G, H)** Detection and quantification of interphase cells with centriole-positive centrosome amplification. (HCT116 (E), CCD 841 CoN (F), RKO (G), and HCT-15 cells (H) were cultured and treated as in (A, B, C, D), and centriole-positive centrosome amplification was detected by immunofluorescence microscopy. Representative images of cells with or without amplified centrosomes are shown (centrioles, Cep135, red; centrosomes, γ-tubulin, green; nuclei, Hoechst 33342, blue; scale bar, 10 μm). Insets show enlarged γ-tubulin, Cep135, or merged signals. The graphs show the quantification of the amount of cells with more than two centrosomes with co-localized γ-tubulin and Cep135 signals (mean ± s.d., n = 3 with a total of 600 cells (E), n = 3 with a total of 1,200 cells (F), and n = 4 with a total of 800 cells (G, H)). Wald's z-statistics computed by the R function *glmmTMB* was used to calculate the *P*-value. \**P* < 0.05; \*\**P* < 0.01; \*\*\**P* < 0.001; and \*\*\*\**P* < 0.0001. A detailed description of statistics is provided in the Materials and Methods section.

*P*-values are available for this figure.

more than two centrioles. This pathway may lead to cells with amplified centrosomes after passage through the next cell cycle. To investigate whether (xeno)estrogens trigger this pathway in our colon cell system, we examined centrosomal levels of Sas-6, which is involved in the initiation of centriole duplication (43), in S phase–synchronized HCT116 and HCT-15 cells (Figs 2E and S2A). Indeed, Sas-6 fluorescence intensities markedly increased in both colon cancer cell lines in response to 10 nM E2, BPA, and DES (Fig 2F and G). Of note, we did not observe an obvious increase in centriole overduplication in the presence of 1 μM of estrogens (Fig S2E and F). The different outcomes are in line with non-monotonic concentration–effect relationships observed for hormones, in which increasing doses do not result in increased effects across the entire concentration range (44, 45).

Because (xeno)estrogens predominantly induced the formation of an odd number of centriole-positive centrosomes (Fig 2A–C),

premature centriole disengagement during early mitosis could be an alternative or parallel route to CA under (xeno)estrogen treatment. If so, we would expect the disengaged centrioles to segregate unevenly during mitosis, with one daughter cell inheriting three centrioles and the other a single centriole (Fig S2B). These split centrosomes may promote the formation of extra spindle poles in the next cell cycle. To test this hypothesis, we released G1/S-synchronized HCT116 and HCT-15 cells in G2 phase, after treatment with (xeno)estrogens until metaphase (Fig S2B). Because the levels of parental centrioles are low in metaphase and signals are not structured (46), we exclusively labeled mitotic centrioles with the more stable procentriole marker Sas-6 (47). Analysis of metaphase cells surprisingly revealed that both opposing centrosomes had a Sas-6 signal that coincides with γ-tubulin in all cases (Fig S2C and D). These results suggest that centrioles segregate evenly between daughter cells and that

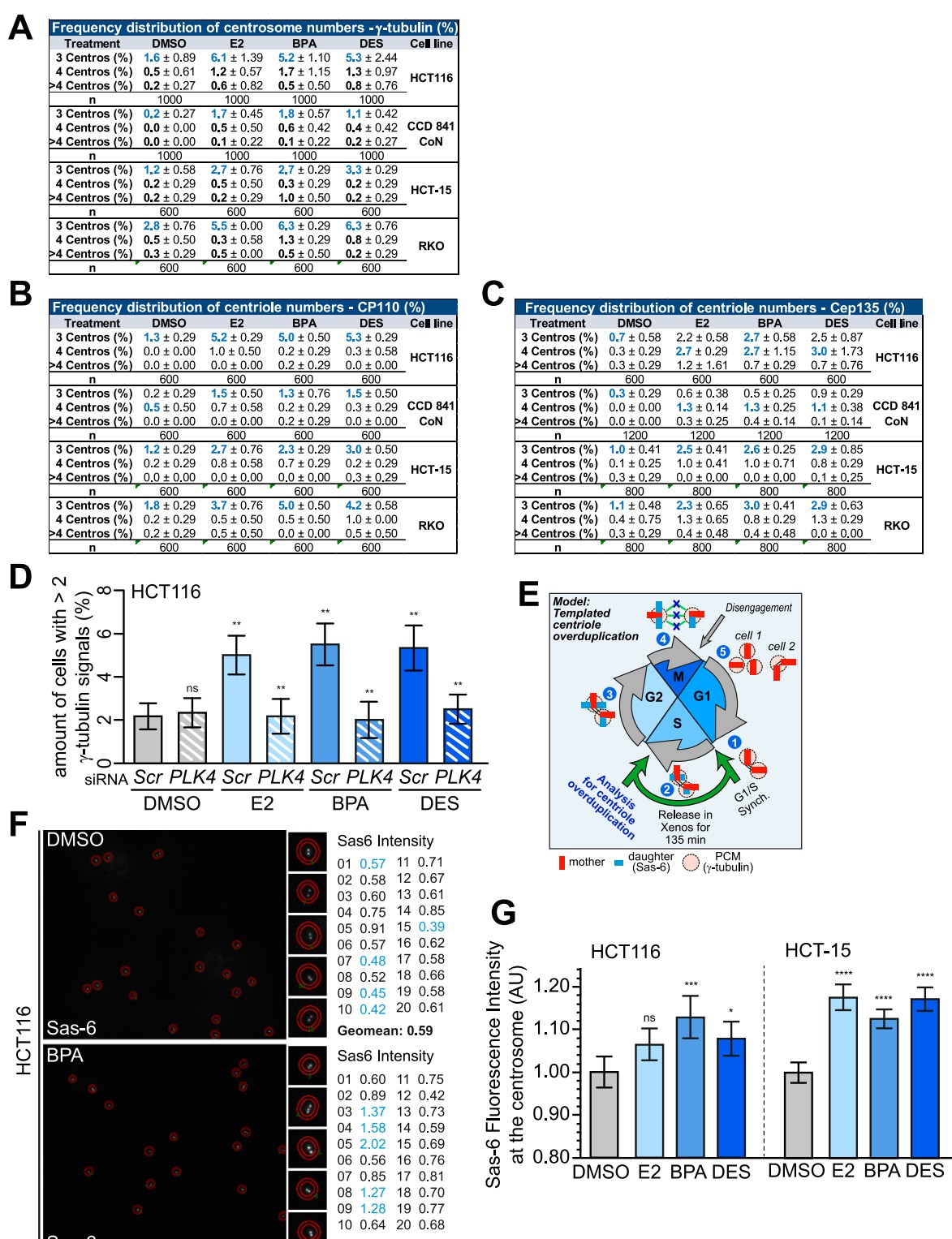

**Figure 2. (Xeno)estrogens trigger centriole overduplication.**
**(A, B, C)** Frequency distribution of centrosome or centriole numbers of (xeno)estrogen-treated HCT116, CCD 841 CoN, HCT-15, and RKO cells. Cells were cultured in a stripped FCS medium and treated with DMSO, or 10 nM 17β-estradiol (E2), bisphenol A (BPA), or diethylstilbestrol (DES) for 48 h. The tables show the quantification of the amount of cells with 3, 4, or more than 4 γ-tubulin (A), CP110 (B), or Cep135 (C) signals at interphase centrosomes derived from Figs 1 and S1H–K. (mean ± s.d., n = 5 with a total of 1,000 cells [(A), HCT116 and CCD 841 CoN] and n = 3 with a total of 600 cells [(A), HCT-15 and RKO]; n = 3 with a total of 600 cells [(B)]; and n = 3 with a total of 600 cells [(C), HCT116] or 1,200 cells [(C), CCD 841 CoN] or 800 cells [(C), HCT-15 and RKO]. **(D)** Quantification of the amount of cells with more than two γ-tubulin signals at interphase centrosomes upon the repression of PLK4 in HCT116 cells and concomitant treatment with E2, BPA, or DES (mean ± s.d., n = 3 with a total of 600 cells).

premature centriole disengagement is quite unlikely. Our data rather support a "templating model" for the extra centrosomes that arise in the presence of (xeno)estrogens. Whether this model explains the generation of predominantly three centrosomes needs to be investigated in future studies.

Collectively, these results support a model in which nanomolar concentrations of E2, BPA, and DES provoke centriole overduplication in colon cancer cells by pushing the parental centrioles to assemble more than one centriole each.

## Estrogen-induced CA depends on GPER1 functionality

The classical estrogen receptors ER$\alpha/\beta$ are not at all or only little expressed in CRC (23, 24, 25, 26, 27). However, several studies demonstrate the alternate estrogen receptor GPER1 being activated in CRC cells upon binding of E2 or BPA (22, 29, 48, 49). In line with these studies, we could detect *GPER1* but not classical estrogen receptor expression on protein levels in all colon (cancer) cell lines studied (Fig S3A). Similarly, the expression of known ER target genes (50) was not affected in the presence of E2, BPA, or DES (Fig S3B), and only HCT116 cells, which ectopically express the classical estrogen receptors, seem to be at least in part responsive for E2-induced *CTSD* expression (Fig S3C). Hence, we suggested a role of GPER1 in estrogen-triggered CA. In line with this reasoning, we partially repressed *GPER1* via gene-specific RNAi (Fig S3D and E) or blocked its activity in response to estrogens, using selective GPER1 antagonists (i.e., G15 and G36 (51, 52)). Subsequently, we checked for CA after additional treatment with estrogens. Indeed, we found a reduced amount of supernumerary centrosomes in estrogen-treated CRC and normal colon epithelial cells upon partial *GPER1* knockdown (Figs 3A and B and S3I). The dependency on GPER1 and the specificity of the *GPER1* knockdown were further emphasized by the result that the co-transfection of a siRNA-resistant version of *GPER1* (RES) restored the (xeno)estrogen response (Fig S3F–H). Similarly, the inhibition of GPER1 suppressed estrogen-induced CA in normal colon and CRC cell lines (Figs 3C and D and S3K). Vice versa, the activation of GPER1 using the specific agonist G-1 (53) or using the anti-estrogens and well-accepted GPER1 activators, tamoxifen and ICI182,780 (22), was sufficient to induce CA (Figs 3E and F and S3J–L). Of note, neither stimulation of GPER1 with E2, BPA, DES, or G1, nor concomitant inhibition with G15 had any apparent effect on colon and CRC cell proliferation within 7 d of treatment (Fig 3G and H). Thus, the supportive role of GPER1 in CRC cell proliferation shown by

others (29, 48, 54) does not appear to be causative for the induction or suppression of centrosome number abnormalities after GPER1 activation or inhibition. Vice versa, the induction of low levels of CA (max 10%; Figs 1 and S1) does not seem to be sufficient to induce a growth disadvantage. This can be observed in p53-proficient cells with high CA (>85–100%, (55)). Together, our results strongly suggest an essential role of GPER1 in the regulation of centrosome numbers in normal colon and CRC cells after treatment with distinct estrogenic GPER1 activators that seem to be independent of GPER's role in CRC proliferation.

## GPER1-activating estrogens induce transient multipolar mitoses and lagging chromosomes

Supernumerary centrosomes are associated with the formation of transient multipolar spindles leading to the formation of lagging chromosomes during anaphase (3). This is in turn an important and well-accepted mechanism for w-CIN and aneuploidy, at least in CRC (6). Because CRC represents a prime example of a tumor entity exhibiting w-CIN (9, 10) with CA as a major underlying molecular mechanism (1, 2), we next tested a potential link between exposure to estrogenic substances, lagging chromosomes, and w-CIN in colon and CRC cell lines. First, we followed up on studies showing that doubling the centrosome number in normal human cells or treating cervical cells with BPA perturbs mitotic progression (15, 56). To this end, we analyzed E2-, BPA-, and DES-treated HCT116 CRC cells expressing GFP-tagged histone H2B in time-lapse microscopy experiments (Figs 4A and S4A and C and Video 1, Video 2, Video 3, Video 4, Video 5, Video 6, Video 7, and Video 8). We found that (xeno)estrogen-treated cells formed multipolar mitoses (Figs 4A, stars; and S4D) but still progressed through mitosis and segregated their chromosomes, although alignment defects, reminiscent of a "pseudo-metaphase" (57), were detected (Figs 4A and S4C, arrowheads; and S4E). We next examined whether cells that exhibit multipolar mitotic figures and chromosome alignment defects are those with supernumerary centrosomes (3). To this end, we requantified these phenotypes in mitotically synchronized cells using adequate markers for centrosomes, mitotic spindles, and chromosomes (Fig 4C and D). The results verified the data derived from H2B-imaged movies (Fig S4D and E). Note that low concentrations of the *Vinca alkaloid* nocodazole did not induce multipolarity as expected but did induce defects in chromosome alignment (58). Surprisingly, the overall duration of prometaphase

---

ANOVA was used to calculate the *P*-value of DMSO + *PLK*4 siRNA. Wald's z-statistics computed by the R function *glmmTMB* was used to calculate the *P*-value of the remaining treatments. ns, not significant; **$P$ < 0.01. **(E)** Graphical scheme for "templated centriole overduplication" (experimental design and hypothesized outcome). Cells were synchronized at G1/S after release in DMSO, or 10 nM E2, BPA, or DES for 135 min. S-phase cells were fixed and stained with markers for γ-tubulin and Sas-6 to visualize centrosomes and daughter centrioles, respectively. Fluorescence intensities of Sas-6 at S-phase centrosomes were measured using the CellProfiler software (F). Centriole overduplication originates from parental centrioles (1) templating the assembly of more than 1 daughter centriole during S phase (upper centrioles panel, [2]), which elongates in G2 (3) and segregates to the mitotic spindle pole (left centrioles panel at metaphase, [4]). After disengagement of centrioles during late mitosis, centrioles split apart, thereby generating daughter cells with more than two centrioles (cell 1). See a detailed description in the Materials and Methods section. **(F)** Shown are maximum projections from z-stacks of representative HCT116 cells treated with DMSO or 10 nM BPA as described in (E). Sas-6 fluorescence intensities (inner circles) were normalized to γ-tubulin and background-corrected (outer circle) using the CellProfiler software. Insets show Sas-6 signals at higher magnification with the corresponding values (highlighted in blue) on the right side. Scale bar, 10 μm. **(G)** Fluorescence intensity of SAS-6 was quantified and plotted from (F). Geometric mean ± 95% CI, n = 3 with a total of 681 cells (HCT116) or 676 cells (HCT-15) from three independent experiments. Mann–Whitney's test was used to calculate the *P*-value. ns, not significant; *$P$ < 0.05; ***$P$ < 0.001; and ****$P$ < 0.0001. A detailed description of statistics is provided in the Materials and Methods section. *P*-values are available for this figure.

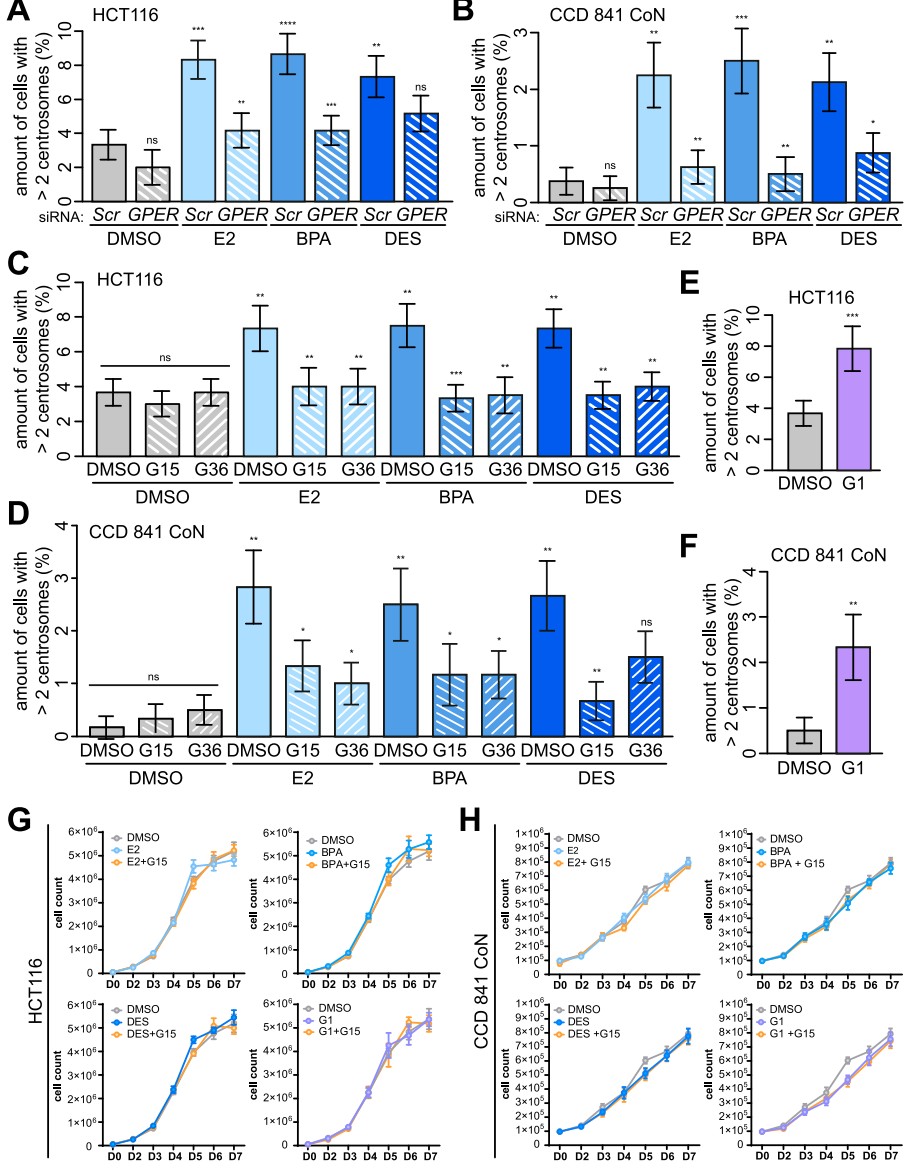

**Figure 3. Centrosome amplification depends on activated GPER1 without effects on cell proliferation.**

**(A, B)** Quantification of centrosome amplification upon *GPER1* knockdown. **(A, B)** HCT116 (A) and CCD 841 CoN cells (B) were transiently transfected with *SCRAMBLED* (Scr) or *GPER1*-specific siRNA (GPER) following treatment with DMSO, or 10 nM 17β-estradiol (E2), bisphenol A (BPA), or diethylstilbestrol (DES) for 48 h. The amount of interphase cells with more than two γ-tubulin signals at centrosomes was quantified (mean ± s.d., n = 3 with a total of 600 cells (A) and n = 4 with a total of 800 cells (B)). ANOVA was used to calculate the *P*-value of DMSO + *GPER1* siRNA. Wald's *z*-statistics computed by the R function *glmmTMB* was used to calculate the *P*-value of the remaining treatments. **(C, D)** Quantification of centrosome amplification upon GPER inhibition. HCT116 (C) and CCD 841 CoN cells (D) were pretreated with 100 nM G15 or G36 for 30 min before additional exposure to DMSO, or 10 nM E2, BPA, or DES for 48 h. The amount of interphase cells with more than two γ-tubulin signals at centrosomes was quantified (mean ± s.d., n = 3 with a total of 600 cells). ANOVA was used to calculate the *P*-value of E2 + G15 and G36. Wald's *z*-statistics computed by the R function *glmmTMB* was used to calculate the *P*-value of the remaining treatments. **(E, F)** Quantification of centrosome amplification upon GPER activation. HCT116 (E) and CCD 841 CoN cells (F) were treated with 100 nM G-1 for 48 h, and the amount of interphase cells with more than two γ-tubulin signals at centrosomes was quantified (mean ± s.d., n = 3 with a total of 600 cells. Values for the DMSO control in (F) are the same as for DMSO treatment in Fig S2I). Wald's *z*-statistics computed by the R function *glmmTMB* was used to calculate the *P*-value. **(A, B, C, D, E, F)** ns, not significant; *P < 0.05; **P < 0.01; ***P < 0.001; and ****P < 0.0001. **(G, H)** Proliferation assay in the presence of DMSO, or 10 nM E2, BPA, or DES for 7 d, with or without 30-min pretreatment with 100 nM G15. 5 × 10⁴ HCT116 (G) and 1 × 10⁵ CCD 841 CoN cells (H) were seeded per six-well plates and manually quantified every day using a hemacytometer and by trypan blue exclusion of dead cells (mean and error ± SEM, n = 4 for E2, BPA, and DES panels, n = 3 for G1 panel (G), and n = 5 for all treatments (H)). A detailed description of statistics is provided in the Materials and Methods section.
*P*-values are available for this figure.

(from nuclear envelope breakdown [NEB] to the onset of anaphase) was not prolonged in the presence of (xeno)estrogens. Thus, the median values of NEB to anaphase onset were not significantly different from those of control-treated cells (Fig 4B). However, treated cells exhibited much greater temporal variability than control cells, with cells spending up to 140 min in a pseudo-metaphase condition (Fig 4B). Therefore, the amount of cells with temporal variability greater than 1.5-fold that of control cells (≥33 min) increased threefold upon (xeno)estrogen treatment and was equivalent to the level of cells exposed to low concentrations of nocodazole (Fig S4B). Of note, nocodazole has previously been shown to increase the mitotic duration in response to unaligned chromosomes (58).

Collectively, our results strengthen the correlation between (xeno)estrogens and supernumerary centrosomes that form transient multipolar mitotic spindles, which resolve during further mitotic progression and initially cause a strong mitotic delay.

Strikingly, chromosome alignment defects frequently accompanied the formation of chromosome bridges (Fig S4C, hash and S4F) and chromosome laggards during anaphase (Fig S4C, arrows). This prompted us to examine whether transient multipolarity after (xeno)estrogen treatment and chromosome alignment defects drive the formation of lagging chromosomes, as expected (3, 59). We therefore synchronized HCT116, HCT-15, and RKO in the anaphase of mitosis using a double thymidine block (Fig S4G) and determined the amount of cells with lagging chromosomes. In fact, we observed an increase in the generation of anaphase lagging chromosomes in all CRC cell lines treated with 10 nM E2, BPA, and DES (Figs 4E and S4H and I). Because we observed CA also in normal colon cells (Fig 1B), we expected that CCD 841 CoN cells also form lagging chromosomes in the presence of (xeno)estrogens. Normal colon cells were more difficult to synchronize in mitosis, so we left them to grow asynchronously in a stripped FCS medium containing the

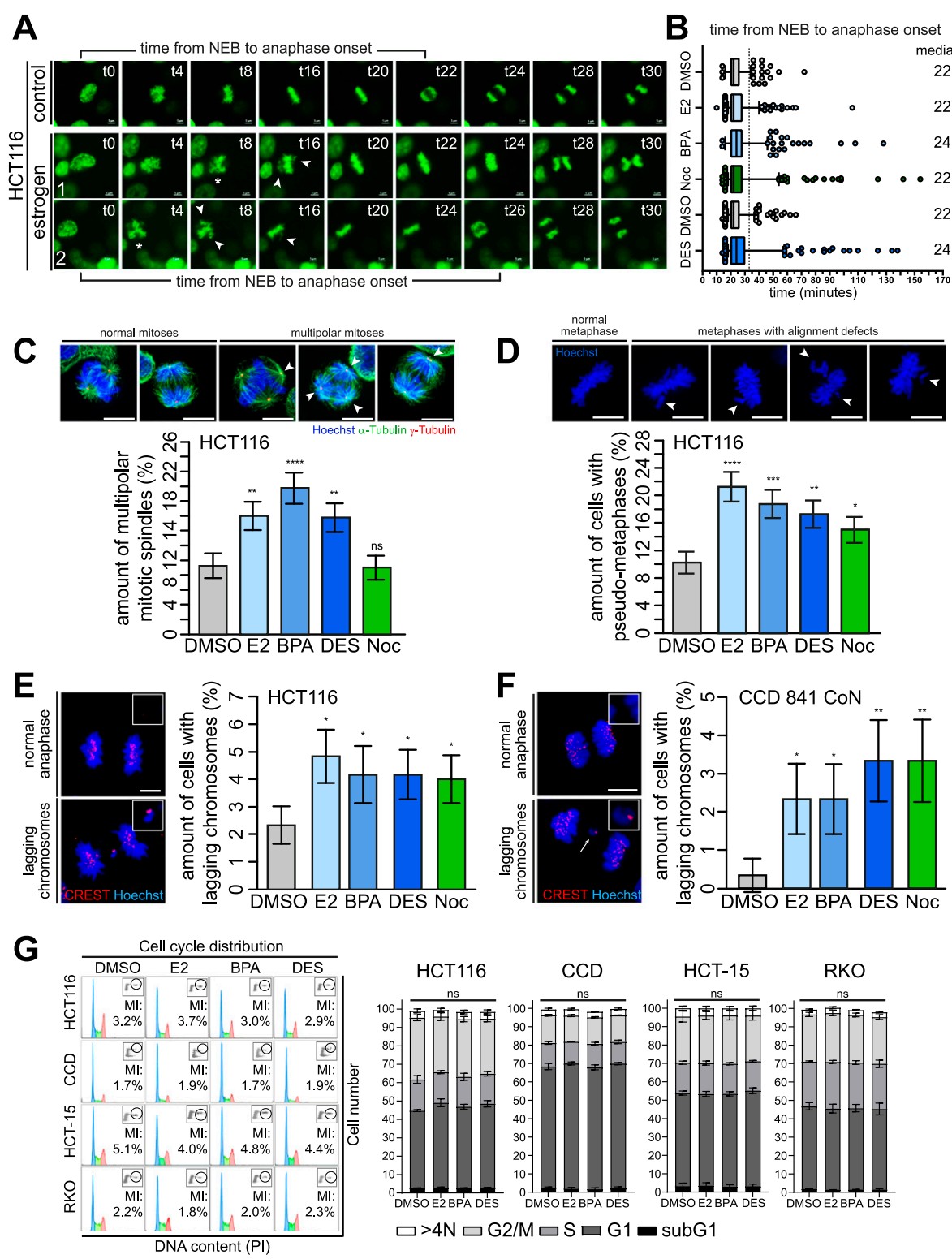

**Figure 4. GPER1-activating estrogens induce multipolar mitoses and lagging chromosomes without altering cell cycle distribution.**
**(A, B)** Disturbance of mitotic progression by treatment with (xeno)estrogens. **(A)** HCT116 cells expressing GFP-tagged histone H2B were treated with DMSO, or 10 nM 17β-estradiol, bisphenol A, or diethylstilbestrol, or 5 nM nocodazole (Noc) for 40 h after live-cell imaging for 8 h under continuous treatment. Still frames were shown from time-lapse movies of representative cells treated with DMSO, bisphenol A (#1), or diethylstilbestrol (#2). Images were captured every 2 min to monitor mitotic progression. Stars point to cells with multipolar chromosome arrangement, and arrowheads, to unaligned chromosomes (t = time in minutes). Scale bars, 5 μm. **(B)** Time from nuclear envelope breakdown to anaphase onset was determined from (xeno)estrogen-treated (10 nM each) and Noc-treated (5 nM) cells (median, box and whiskers, 5–95 percentile, n = 4 with a total of 400 cells). **(C, D)** Detection and quantification of HCT116 cells with multipolar spindles (C) and pseudo-metaphases (D). Cells treated as in (A)

GPER1 ligands for 48 h and checked for the formation of mitotic laggards in anaphase (Fig 4F). Consistent with CRC cells, non-transformed CCD 841 CoN also showed an increase in the formation of lagging chromosomes after exposure to E2, BPA, and DES (Fig 4F). As expected, non-transformed cells had an overall lower number of cells with mitotic laggards, but with much higher fold induction values after (xeno)estrogen treatment. Of note, the overall level of (xeno)estrogen-induced lagging chromosomes was comparable to treatment with the w-CIN inducer nocodazole in both HCT116 and CCD 841 CoN. Importantly, neither (xeno)estrogen-induced CA nor mitotic laggard formation appeared to affect cell cycle distribution, as the percentages of colon and CRC cell lines treated with DMSO, E2, BPA, and DES remained almost the same with respect to G1, S, G2, and M phases (Fig 4G). These results suggest that the low nanomolar concentrations of E2, BPA, and DES used in this study are sufficient to induce mild mitotic defects that do not disrupt mitotic progression to a large extent, that is, which would be associated with cytotoxicity, but are sufficient to cause defects in chromosome segregation that could manifest in w-CIN and genome instability.

### GPER1-activating estrogens induce w-CIN

Because lagging chromosomes represent a direct precursor of w-CIN (3, 6), estrogen exposure was expected to similarly increase levels of w-CIN and aneuploidy. To address this hypothesis and to avoid clonal effects, we generated different single-cell clones derived from parental HCT116 cells and analyzed the evolution of karyotypes in the presence of 10 nM E2, BPA, DES, or nocodazole within a defined time span of 30 generations. Indeed, estrogen-treated cell clones became aneuploid (Fig 5A) and evolved an increase in karyotype variability to a level that was similar to nocodazole, while maintaining the same modal number of chromosomes (Figs 5B and S5A). Similar effects were observed for HCT-15 CRC cells permanently treated with the estrogens (Fig S6F, colored, non-patterned bars; and S6G). Importantly, chromosome counting from metaphase spreads and interphase FISH analyses revealed w-CIN and aneuploidy even in normal colon epithelial cells with an overall induction that was comparable to nocodazole (Figs 5C and D and S5B and C). Thus, our data suggest that exposure to GPER1-activating estrogens, such as E2, BPA, and DES, may promote genomic instability in intestinal cells that could persist during CRC progression. Of note, chromosomally instable cells used for karyotype analysis after long-term treatment with (xeno)estrogens (Figs 5A–D, S5A–C, and S6F, DMSO) exhibited supernumerary centrosomes (Figs 5E and F and S5F). Strikingly, similar to the short-term exposure (Figs 1 and S1D and E), the total CA content was below 10% and cells had mainly three PCM-containing centrosomes (Fig S5D, E, and G). These results not only demonstrate a direct link between (xeno)estrogen-induced CA and w-CIN (3), which has not yet been demonstrated in a colon (cancer) system. Our data also imply that the generation of just one extra centrosome per cell and overall low frequencies of CA are sufficient for w-CIN development. In contrast, large numbers of supernumerary centrosomes per cell with high frequencies are likely to adversely affect cell viability because extra centrosomes cluster inefficiently during mitosis and increase the frequency of lethal multipolar divisions (3).

### Estrogen-induced genomic instability depends on GPER1 functionality

Given that our data demonstrate an essential role of GPER1 in estrogen-triggered CA (Fig 3), we reasoned that the formation of lagging chromosomes (Fig 4) and w-CIN (Fig 5) is also dependent on the alternate estrogen receptor. Our hypothesis of GPER1 dependence of (xeno)estrogen-triggered mitotic laggards proved to be correct, because the partial repression of *GPER1* with gene-specific siRNAs resulted in a significant reduction in lagging chromosomes in HCT116 upon E2 or BPA treatment (Fig 6A). At least the same tendency was observed upon DES treatment. The dependency on GPER1 and the specificity of the *GPER1* knockdown were confirmed by the co-transfection of an siRNA-resistant version of *GPER1* (RES), which restored the (xeno)estrogen-triggered response (Fig S6A). GPER1 dependency was also observed in normal colon cells after gene-specific knockdown followed by estrogen treatment (Fig S6B). Obviously, this cell system was pushed to its limit by this experimental setup, indicated by the small number of anaphase cells that could be evaluated per condition. Therefore, we decided to repeat the evaluation of lagging chromosomes using the GPER1-selective antagonist G15. Consistently, the inhibition of GPER1 by pretreatment of normal colon (Fig 6B) and CRC cell lines (Fig S6C and D) with G15 before (xeno)estrogen exposure restored the amount of anaphase cells with laggards to control levels. As expected, the formation of lagging chromosomes induced by nocodazole did not

---

were synchronized in mitosis with a double thymidine block as described in (9). Representative immunofluorescence images show cells with or without multipolar mitotic spindles (C) or chromosome alignment defects (D) (chromosomes, Hoechst 33342, blue; centrosomes, γ-tubulin, red; spindles, α-tubulin, green; scale bar, 10 μm). Arrowheads mark extra centrosomes (C) and unaligned chromosomes of pseudo-metaphases (D). The graphs show the quantification of the proportion of cells exhibiting the respective mitotic defect as indicated (mean ± s.d., n = 4 with a total of 400 cells). Wald's z-statistics computed by the R function *glmmTMB* was used to calculate the P-value. ns, not significant; (*P < 0.05; **P < 0.01; ***P < 0.001; and ****P < 0.0001). **(E, F)** Detection and quantification of anaphase cells with lagging chromosomes. Colon (cancer) cells were treated as in (A) and synchronized in the anaphase of mitosis with a double thymidine block as described in (9) (HCT116, (E)) or left grown asynchronously for 48 h (CCD 841 CoN, (F)). Representative immunofluorescence images with or without lagging chromosomes are shown (chromosomes, Hoechst 33342, blue; kinetochores, CREST, red; scale bar, 10 μm). Insets show lagging chromosomes at higher magnification. Only kinetochore-positive chromosomes were counted as lagging chromosomes (arrows). Graphs show the quantification of the proportion of cells exhibiting lagging chromosomes (mean ± s.d., (D) n = 6 with a total of 600 cells, Wald's z-statistics computed by the R function *glmmTMB* was used to calculate the P-value, and (E) n = 3 with a total of 300 cells). The bootstrap procedure was used to calculate the P-value. (*P < 0.05 and **P < 0.01). **(G)** Representative FACS histograms (left) of HCT116, CCD 841 CoN (CCD), HCT-15, and RKO cells treated as in (A) for 48 h showing cell cycle distribution and mitotic indices (MI) of propidium iodide and MPM2–co-immunostained cells. Blue, G1 phase; green, S phase; and red, G2 phase. The graphs (right) show the quantification of the amount of cells in the subG1 area, and G1, S, or G2 phase based on their DNA content (N). >4N = polyploid cells (mean ± SEM, n = 3 with a total of 30,000 cells, ordinary one-way ANOVA). A detailed description of statistics is provided in the Materials and Methods section. P-values are available for this figure.

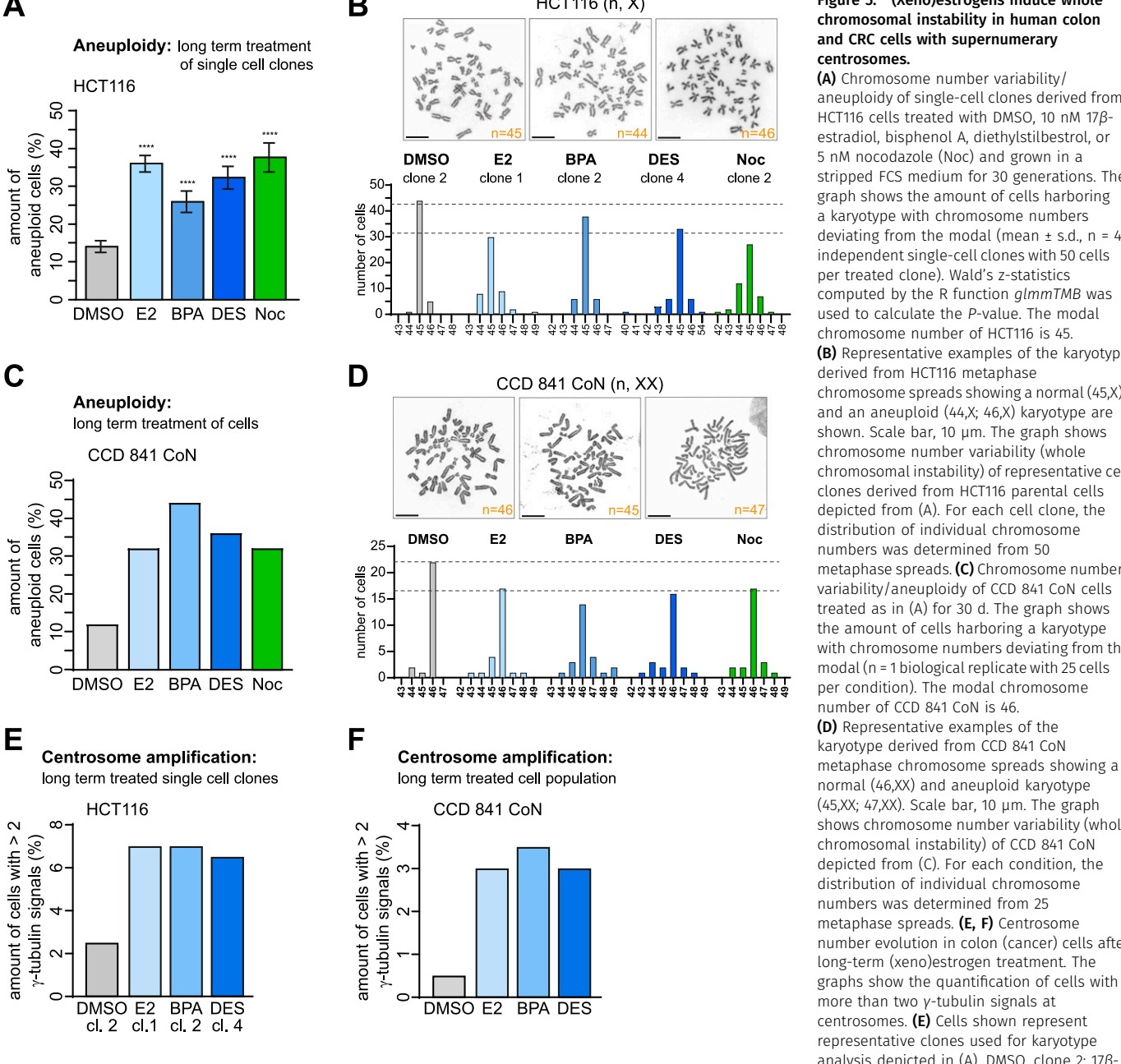

**Figure 5. (Xeno)estrogens induce whole chromosomal instability in human colon and CRC cells with supernumerary centrosomes.**
**(A)** Chromosome number variability/aneuploidy of single-cell clones derived from HCT116 cells treated with DMSO, 10 nM 17$\beta$-estradiol, bisphenol A, diethylstilbestrol, or 5 nM nocodazole (Noc) and grown in a stripped FCS medium for 30 generations. The graph shows the amount of cells harboring a karyotype with chromosome numbers deviating from the modal (mean ± s.d., n = 4 independent single-cell clones with 50 cells per treated clone). Wald's z-statistics computed by the R function *glmmTMB* was used to calculate the *P*-value. The modal chromosome number of HCT116 is 45. **(B)** Representative examples of the karyotype derived from HCT116 metaphase chromosome spreads showing a normal (45,X) and an aneuploid (44,X; 46,X) karyotype are shown. Scale bar, 10 μm. The graph shows chromosome number variability (whole chromosomal instability) of representative cell clones derived from HCT116 parental cells depicted from (A). For each cell clone, the distribution of individual chromosome numbers was determined from 50 metaphase spreads. **(C)** Chromosome number variability/aneuploidy of CCD 841 CoN cells treated as in (A) for 30 d. The graph shows the amount of cells harboring a karyotype with chromosome numbers deviating from the modal (n = 1 biological replicate with 25 cells per condition). The modal chromosome number of CCD 841 CoN is 46. **(D)** Representative examples of the karyotype derived from CCD 841 CoN metaphase chromosome spreads showing a normal (46,XX) and aneuploid karyotype (45,XX; 47,XX). Scale bar, 10 μm. The graph shows chromosome number variability (whole chromosomal instability) of CCD 841 CoN depicted from (C). For each condition, the distribution of individual chromosome numbers was determined from 25 metaphase spreads. **(E, F)** Centrosome number evolution in colon (cancer) cells after long-term (xeno)estrogen treatment. The graphs show the quantification of cells with more than two γ-tubulin signals at centrosomes. **(E)** Cells shown represent representative clones used for karyotype analysis depicted in (A). DMSO, clone 2; 17β-estradiol, clone 1; bisphenol A, clone 2; and diethylstilbestrol, clone 4 (mean, n = 1 biological replicate with 200 cells per condition). **(F)** Cells analyzed were derived from the cell population used for karyotype analysis depicted in (C) (mean, n = 1 biological replicate with 200 cells per condition). A detailed description of statistics is provided in the Materials and Methods section.
*P*-values are available for this figure.

depend on *GPER1* expression or activity (Fig 6A and B). These results reinforce the role of estrogen-activated GPER1 in lagging chromosome formation, while simultaneously suggesting an involvement of the receptor in w-CIN and aneuploidy. To investigate a possible link between GPER1 and w-CIN, we generated single-cell clones derived from HCT116 parental cells stably expressing shRNAs targeting *GPER1* and exposed them to estrogens for 30 generations (Fig S6E). We showed that these single-cell clones failed to evolve w-CIN and aneuploidy upon estrogen exposure (Fig 6C and E). In

contrast, the karyotype stability of nocodazole-treated cell clones was not restored to control levels, indicating a specific role of GPER1 in w-CIN in the presence of distinct estrogens. Importantly, we could validate these results obtained from HCT116 in another CRC cell line, that is, HCT-15, and also in non-transformed CCD 841 CoN colon epithelial cells after the inhibition of GPER1 by the selective GPER1 antagonist G15 (Figs 6D and F and S6F and G). Overall, our results describe a molecular mechanism linking distinct estrogenic substances and GPER1

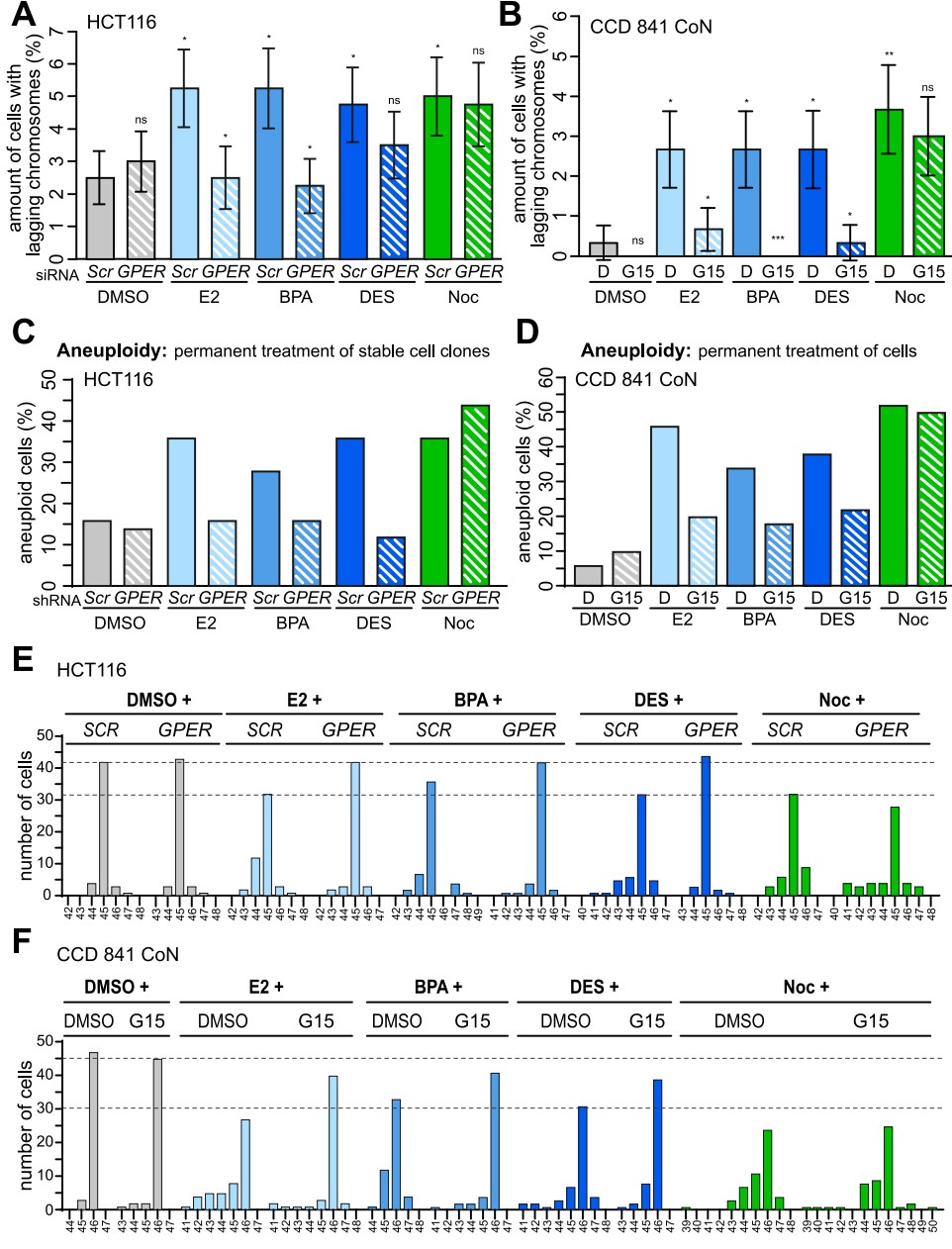

**Figure 6. Mitotic laggards and whole chromosomal instability depend on estrogen-activated GPER1 function.**
**(A)** HCT116 cells were transiently transfected with *SCRAMBLED* (*Scr*) or *GPER1*-specific siRNA (*GPER*) after treatment with DMSO, 10 nM 17β-estradiol (E2), bisphenol A (BPA), diethylstilbestrol (DES), or 5 nM nocodazole (Noc) and synchronization in the anaphase of mitosis as described in (9). The quantification of the amount of anaphase lagging chromosomes is shown (mean ± s.d., n = 4 with a total of 400 cells). ANOVA was used to calculate the *P*-value of DMSO + *GPER1* siRNA. Wald's z-statistics computed by the R function *glmmTMB* was used to calculate the *P*-value of the remaining treatments. ns, not significant; *$P < 0.05$ and **$P < 0.01$. **(B)** CCD 841 CoN cells pretreated with G15 for 30 min after additional treatment as in (A) for 48 h (median ± s.d., n = 3 with a total of 300 cells). ANOVA was used to calculate the *P*-value of DMSO + *GPER1* siRNA and all G15 treatments. The bootstrap procedure was used to calculate the *P*-value of the remaining treatments. ns, not significant; *$P < 0.05$; **$P < 0.01$; and ***$P < 0.001$. **(C)** HCT116 single-cell clones stably expressing *SCRAMBLED* (*Scr*) or shRNAs targeting *GPER1* (*GPER*) were grown for 30 generations in DMSO, 10 nM E2, BPA, DES, or 5 nM Noc. The graph shows the amount of cells harboring an aneuploid karyotype with chromosome numbers deviating from the modal (modal number = 45). Data were collected from 50 cells per clone (n = 1 biological replicate). **(D)** CCD 841 CoN cells pretreated with G15 for 30 min before additional exposure to DMSO, 10 nM E2, BPA, DES, or 5 nM Noc for 30 generations. The graph shows the amount of cells (N = 50) harboring an aneuploid karyotype with chromosome numbers deviating from the modal (modal number = 46); (n = 1 biological replicate). **(E)** Chromosome number variability/whole chromosomal instability of HCT116 cell clones depicted from (C). For each cell clone, the distribution of individual chromosome numbers was determined from 50 metaphase spreads. **(F)** Chromosome number variability/whole chromosomal instability of CCD 841 CoN cells depicted from (D). For each condition, the distribution of individual chromosome numbers was determined from 50 metaphase spreads. A detailed description of statistics is provided in the Materials and Methods section. *P*-values are available for this figure.

## Discussion

Over the past years, the evidence of estrogen actions in CRC has been accumulating, suggesting that the colon, similar to the classic hormonally regulated tissues (i.e., breast, ovary, and prostate), is an important estrogen-regulated tissue. Orally ingested estrogens come into direct contact with the gastrointestinal tract, where they might exert their specific effects (28, 60). Although a slightly lower CRC incidence and mortality in women compared with men suggests a protective role of estrogens against CRC, on the one hand (28), the experimental evidence, on the other hand, demonstrates pathophysiological effects of estrogenic substances on the colon. Specifically, oral contraceptives and hormone therapy were shown to be associated with an increased risk of inflammatory bowel disease in (postmenopausal) women (61, 62), and colitis-associated cancer in ovariectomized mice supplemented with estrogens (63). Further studies demonstrated that industrial estrogens, such as DES and BPA, increase the incidence of chemically induced colon

with numerical centrosome defects and w-CIN in non-trans-formed colon and CRC cell systems.

carcinoma (31, 32) or promote migration, invasion, and metastasis of CRC cells through the modulation of protein profiles and the induction of epithelial-to-mesenchymal transition (12, 64). Together, experimental results strongly suggest a role of distinct (xeno)estrogens in the development of gastrointestinal diseases.

Our study has uncovered a novel role of E2, BPA, and DES in a colon cell system, involving numerical centrosome defects at low nanomolar concentrations sufficient to cause the formation of lagging chromosomes that manifest in w-CIN and aneuploidy. W-CIN represents a major route to CRC (10) with numerical CA being a widespread lesion in colon carcinogenesis (1, 2). Indeed, colon carcinoma bears significantly higher centrosome numbers per cell than adenoma or normal colonic epithelium, and CA is associated with higher histologic grades of dysplastic and invasive lesions (1, 65). Our results hypothesize a potential cause of amplified centrosomes and w-CIN in CRC cells that may include exposure to certain environmental estrogenic substances.

Importantly, we propose a molecular underlying mechanism for this estrogen-triggered aneugenic effect in colon cells, which includes the formation of lagging chromosomes and involves the alternate estrogen receptor GPER1 (Figs 6A and B and S6A–D). Indeed, the siRNA-mediated knockdown of *GPER1* and, more significantly, receptor inhibition suppressed both estrogen-induced CA (Figs 3A–F and S3G–L) and w-CIN (Figs 6C–F and S6F and G). However, there is conflicting evidence about anti- and protumorigenic effects of GPER1 in colorectal carcinogenesis, potentially depending on oxygen levels of cancer cells and the tumor microenvironment (29). Tumor suppressor functions for GPER1 came from studies that showed decreased *GPER1* expression levels in CRC patients and colorectal adenoma tissues, which correlated with increased tumor progression, lymph node metastasis, and decreased survival rates (66). Similarly, activated GPER1 is shown to have a detrimental effect on CRC cell proliferation (66, 67). In contrast, accumulating evidence suggests that GPER1 might act as a tumor promoter in CRC. High *GPER1* expression levels were found to be significantly associated with poor relapse-free survival in women with stage 3 or 4 CRC indicating a role of GPER1 in CRC progression and survival, potentially as a result of estrogen-dependent signaling in CRC (29). Consistently, increased colonic activity of estrogen-activating or estrogen-converting enzymes in CRC patients and tissue samples leading to estrogen-mediated GPER1 activation was associated with increased CRC cell proliferation *via* GPER1 (48). These results support the findings of other studies suggesting GPER1 to mediate CRC cell proliferation upon exposure to GPER1 activators (54), at least under hypoxic conditions (29). We could not observe a role of estrogens or GPER1 in cell proliferation (i.e., under normoxic conditions), but we did show an effect on key CRC lesions, that is, CA and w-CIN that may contribute to tumorigenesis, tumor progression, or therapy resistance (7, 8). Estrogen actions of GPER1 were also shown in both neoplastic transformation of the colon and tumor progression, including colonic motility, immune regulation, and inflammation (referred to in reference 28), further supporting a protumorigenic role of estrogen-activated GPER1. It would be interesting to investigate in future studies whether activated GPER1 would affect clonogenicity or anchorage-independent growth in colon cells as a readout for malignant transformation (68, 69).

The exact mechanism by which GPER1 responds to estrogenic substances remains elusive and cannot be derived solely by the binding affinities of the ligands to the receptor, that is, ranging from (low) nanomolar (e.g., E2 and BPA) to micromolar concentrations (e.g., DES, 4-OH-tamoxifen, and atrazine) (22). In fact, we observed similar overall levels of CA in the presence of the GPER1 activators E2, BPA, and DES (i.e., estrogenic substances) or ICI182,780 and tamoxifen (i.e., anti-estrogens), but only a partial suppression of CA in cells with low *GPER1* expression that were treated with BPA (HCT-15, Fig S3I). These results might suggest different sensitivities of colon (cancer) cells toward distinct estrogenic substances or likely reflect different GPER1-mediated signaling pathways potentially activated by varied estrogenic and anti-estrogenic GPER1 binders. In fact, both ICI182,780 and tamoxifen were shown to increase steroid sulfatase activity in a GPER1-dependent manner, suggesting that these pharmaceuticals induce the conversion of circulating estrogens to the active forms that, in turn, might trigger the specific effect (48). E2, BPA, and DES may promote a centrosome-dependent pathway to CRC *via* GPER1. The latest hints for a link of GPER1 to the centrosome came from studies in rodents and ovarian cancer cells reporting G-1 facilitated the activation of protein kinase A (70) and the phosphorylation of Aurora A kinase (71), respectively. Both centrosome-associated kinases are involved in CA, probably through the phosphorylation and the stabilization of the centriolar protein centrin (72, 73). Further studies will reveal whether our observations follow this axis or whether an alternative route will trigger the estrogen/GPER1-mediated induction of CA in colon cells.

The molecular mechanisms causing numerical centrosome alterations are multifaceted (referred in reference 35, 74), and estrogenic substances appear to play a pivotal role in this process (35). Kim and colleagues (15) demonstrated that BPA causes multipolar mitotic spindles in HeLa cells by inducing centriole overduplication and premature centriole disengagement, that is, in a classical estrogen receptor–independent manner. Although we observed the formation of predominantly three centrosomes (Figs 1 and 2A–C), our data do not clearly support the latter mechanism because the centrioles evenly distribute between both opposite spindle poles in metaphase (Fig S2C and D). In contrast, we have promising results favoring centriole overduplication. First, the down-regulation of *PLK4* restored normal centrosome numbers in the presence of (xeno)estrogens (Fig 2D). Secondly, we found that E2, BPA, and DES cause increased levels of Sas-6 at the centrosome during S phase (Fig 2F and G). This result hypothesizes that parental centrioles template the assembly of more than one daughter centriole each that may form additional centrosomes in the following mitosis (Fig 2E). Whether and how (xeno)estrogen-activated GPER1 might be involved in this pathway and whether other important regulatory proteins of centriole biogenesis such as STIL or the Plk4-recruiting factors Cep192 and Cep152 are also involved (75, 76) remains an interesting question for future studies. Of note, centriole overduplication can also arise from de novo centriole assembly, which is normally driven in the absence of pre-existing centrioles (77). Importantly, sufficiently high levels of cytoplasmic Plk4 can trigger the de novo pathway in human cultured cells, regardless of the presence or absence of pre-existing centrioles (78). It would be interesting to investigate whether cytoplasmic

levels of Plk4 increase in the presence of (xeno)estrogens, thereby triggering the de novo pathway.

Of note, the percentage of cells with CA is low (i.e., <10%, Fig 1) but comparable to data from previous studies detecting ~10––15% of cells with amplified centrosomes after exposure to low estrogen concentrations (14, 15, 21). Even the overexpression of *PLK4*, a key regulator of centrosome duplication (37), causes only a slightly higher proportion of cells with CA (Fig S1E, (3)), suggesting a threshold above which a growth disadvantage of cells with CA should be suspected. Consistent with this hypothesis, a study by Holland et al illustrates a decline in cells with a rapid and highly penetrant CA (>85–95%) to values <10% because of proliferation deficits in the cells harboring extra centrosomes (55). We predict that a larger proportion of cells with CA will be detrimental to long-term survival. This fits with our data illustrating a constant low level of CA below 10% over 30 generations of estrogen treatment (Figs S1C and D, 5E and F, and S5D–G) that did not apparently affect cell cycle distribution (Fig 4G) or cell proliferation (Fig 3G and H). The low hormone concentrations used in this study are important in this context because it has been previously reported that higher doses in the micromolar range suppress microtubule polymerization and dynamics or even disrupt the microtubule network and arrest cells in mitosis, possibly by mechanisms similar to spindle poisons (79, 80, 81, 82, 83, 84). In contrast, nanomolar hormone concentrations did not arrest cells in mitosis (Fig 4G). This is important because mitotic arrest can be lethal in several ways (summarized in reference 85), counteracting continuous chromosome missegregation that manifests in w-CIN and aneuploidy.

Although many causes may underlie the etiology of CA in CRC, the cellular consequences are clear and unambiguous. Supernumerary centrosomes lead to the formation of multipolar spindles, which cause multipolar cell divisions if not corrected otherwise. However, multipolar mitoses are likely to be detrimental to cells because of gross chromosome missegregation and cell death (3). Remarkably, (xeno)estrogens trigger the generation of supernumerary centrosomes and, as expected, the formation of multipolar mitotic spindles (Figs 1 and 4C). However, we did not apparently observe multipolar mitoses and cell death in live-imaged cells that were exposed to (xeno)estrogens (Figs 4A and S4C). Importantly, (cancer) cells can "cope" with extra centrosomes by several mechanisms to limit the detrimental consequences of CA. These include centrosome removal, centrosome inactivation, asymmetrical segregation during cell division, or centrosome clustering (86). We hypothesize that (xeno)estrogen-treated cells likely avoid lethal divisions by coalescence of extra centrosomes to form a pseudo-bipolar spindle. Consistent with this, (xeno)estrogen-treated cells display an increased frequency of lagging chromosomes during anaphase (Figs 4E and F and S4G and H), which typically arise from multipolar spindle intermediates and merotelic kinetochore attachments (3). As seen for a moderate increase in Plk4 (87), (xeno)estrogens mainly induce the creation of just one extra centrosome per cell, which is permissive for centrosome coalescence and cell survival. In contrast, large numbers of supernumerary centrosomes are likely to be detrimental to cell viability because of inefficient clustering before division. Of note, supernumerary centrosomes and lagging chromosomes evolve in both normal colon epithelial and CRC cells, suggesting that centrosome clustering is likely not only restricted to

cancer cells but also possible in non-transformed cells with extra centrosomes, which is in line with previous studies (referred to in reference 86). It would be interesting to follow the fate of extra centrosomes in the absence of the causative trigger, that is, (xeno) estrogens. A recent study by Sala et al demonstrates that centriole numbers return to normal after initial amplification *via* the over-expression of *PLK4* in hTERT-RPE-1 cells (88). The authors argue against centriole elimination or asymmetric segregation, but, consistent with the previous work (55), conclude that cells with extra centrioles were outcompeted by cells in the population with a normal number, because of growth disadvantage. Given that (xeno) estrogens induce prolonged mitosis only in a subset of cells (Figs 4B and S4B), which are most likely those with extra centrosomes (Figs 4C and S4D), but do not apparently affect the proliferation of cells without CA (Fig 3G and H), it seems likely that cells with normal centrosome numbers will overgrow cells with CA over time.

Consistent with the established role of extra centrosomes and lagging chromosomes in driving w-CIN and aneuploidy, we observed supernumerary centrosomes in aneuploid cells, which evolved from long-term treatment with (xeno)estrogens (Figs 5E and F and S5D–G). Of note, aneuploidy varied between 30 and 40% irrespective of cell transformation, as normal colon CCD 841 CoN and CRC-derived HCT116 cells behave similarly (Fig 5A and C). Furthermore, treated cells are chromosomally instable and display ongoing gains and losses of several chromosomes, not least of chromosomes 2 and 8 (Figs 5B and D, S5A–C, and S6G), suggesting complex karyotypes that were triggered by (xeno)estrogens. In this context, it would be interesting to investigate the copy-number heterogeneity of the karyotypes from non-tumor and tumor cell lines in the presence of (xeno)estrogens. By inducing extremely high rates of chromosome missegregation in yeast, Madhwesh et al demonstrate that distinct patterns of complex karyotypes are created over time, with maximized selective advantages of distinct chromosomal aneuploidies (89). Single-cell sequencing studies would therefore offer a reliable method for examining the karyotypes of (xeno)estrogen-treated single-cell clones at high resolution and will give an important impact on how distinct estrogenic substances might influence karyotype diversity. Interestingly, an in silico study by Elizalde et al (90) demonstrates that karyotype diversity is significantly more dependent on the chromosome missegregation rate than on the number of cell divisions. Thus, karyotype diversity can be reached rapidly at high missegregation rates. In contrast, at low missegregation rates karyotype diversity is expected to be constrained after more cell division events. These results are important for our data demonstrating lower rates of lagging chromosomes in CCD 841 CoN (2.5–3%) compared with HCT116 (~5%), serving as indicators for different chromosome missegregation rates (Fig 4E and F). This leads us to speculate that maximal karyotype heterogeneity might not be exhausted in both cell lines after 30 d of (xeno)estrogen treatment, but even more in CCD 841 CoN cells. It would therefore be reasonable to investigate karyotype diversity after further cell division events in the presence of (xeno)estrogens.

Collectively, our results provide the first demonstration that (xeno)estrogens might trigger key cancer-prone lesions and point GPER1 to an important regulatory protein mediating CA, w-CIN, and aneuploidy in a non-classical hormone-regulated colon cell

system. Because the role of GPER1 in colon cancer development is still under debate, future studies are needed demonstrating to what extent centrosomal proteins are involved in the GPER1-mediated aneugenic effect observed in this colon (cancer)-derived study.

# Materials and Methods

### Cell culture, reagents, and treatments

CRC cell lines HCT116, HCT-15, and RKO, and normal colon epithelial cells CCD 841 CoN were acquired from the American Type Culture Collection. CRC cell lines were cultured in phenol red–free RPMI 1640 (PAN-Biotech), and CCD 841 CoN, in DMEM (PAN-Biotech), each supplemented with 10% charcoal-stripped fetal calf serum (Th. Geyer GmbH) and 1% penicillin/streptomycin (Merck Millipore) at 37°C and 95% humidity and 5% CO₂. Culture medium conditions were used for all experiments. All chemicals were purchased from Sigma-Aldrich unless otherwise stated, dissolved in DMSO, and stored at −20°C. The cells were treated with 10 nM E2, 10 nM BPA, 10 nM DES, 5 nM nocodazole (Noc), 10 nM tamoxifen (Tam), 100 nM G15 (Tocris Bioscience), 100 nM G36 (Biomol), 100 nM ICI182,780 (ICI), and 100 nM G-1 (Tocris Bioscience) for 48 h in a stripped medium unless otherwise stated.

### siRNAs and shRNA plasmids

The following siRNA sequences (Qiagen) were used:

*SCRAMBLED* (control): 5′-CAUAAGCUGAGAUACUUCA-3′
*GPER1* no. #1: 5′-GCUGUACAUUGAGCAGAAA(TT)-3′
*GPER1* no. Q2 (CCD 841 CoN): 5′-CTGGATGAGCTTCGACCGCTA-3′

Two oligonucleotides encoding human *GPER1* short hairpin sequence (5′-TGCACTCCTCACACA GAATTGCTACAATC-3′: sense, TF316565D; and 5′-GATTGTAGCAATTCTGTGTGAGGAGTG CA-3′, anti-sense) were synthesized (Eurofins Genomics). The control *SCRAMBLED* shRNA sequence with no target in the human genome was 5′-CAUAAGCUGAGAUACUUCA-3′. To generate a *GPER1* shRNA repression plasmid, pSuper-Retro was digested using HindIII and BglII enzymes (New England Biolabs) according to manufacturer's instructions after ligation and transformation into competent DHα5 E.coli cells (Invitrogen). Cells were cultivated, selected with 100 µg/ml ampicillin, and plated. Colonies were picked and incubated with selection LB medium (Roth) overnight. Plasmids were purified using NucleoBond Xtra Plasmid Purification Kit (Macherey-Nagel) according to the manufacturer's protocol. Plasmids were digested and resolved on a 1.5% agarose gel to ensure proper ligation. Final plasmids were sequenced with Sanger sequencing (Eurofins), and the knockdown of *GPER1* was verified by qRT-PCR. HCT116 cells stably expressing pRetro-Super *SCRAMBLED* and pRetro-Super *GPER1* were used to generate stable cell lines. The *GPER1* siRNA rescue vector pRP-hGPER1-siRES containing silent mutations in the *GPER1* siRNA-targeted sequence (NM_001505.3: codons 335–341 were changed to AGGCTCTATATCGAACAGAAA), was obtained from VectorBuilder.

### RNA extraction and quantitative real-time PCR

Total RNA was extracted with Qiagen RNeasy Mini Kit according to manufacturer's instructions from cells growing in a six-well plate. cDNA was synthesized from 1 µg of total RNA with High-Capacity cDNA Reverse Transcription Kit (Thermo Fisher Scientific) on a QuantStudio 7 Flex PCR machine (Thermo Fisher Scientific) using optimized conditions for PowerUp SYBR Green dye (Thermo Fisher Scientific): holding stage: 50°C for 2 min and 90°C for 10 min; cycling stage: 40 cycles at 95°C for 15 s and 60°C for 1 min; and melt curve stage: 95°C for 15 s followed by 60°C continuous heating to 95°C for 1-min holding at 95°C for 30 s and 60°C for 15 s. The amplification specificity was confirmed by melt curve analysis. A relative gene expression was calculated using ΔΔCt, with GAPDH as a normalization control. The following primer sequences were used for the qRT-PCR:

*CTSD*: F 5′-CTACCTGAATGTCACCCGCA-3′
R 5′-GGGATCATGTACTCGCCCTG-3′
*ERBB2*: F 5′-GGCCGTGCTAGACAATGGAG-3′
R 5′-GGGTTCCGCTGGATCAAGAC-3′
*GAPDH*: F 5′-TGCACCACCAACTGCTTAGC-3′
R 5′-GGCATGGACTGTGGTCATGAG-3′
*GPER1 #1*: F 5′-CGTCATTCCAGACAGCACCGAG-3′
R 5′-CGAGGAGCCAGAAGCCACATC-3′
*GPER1 #2*: F 5′-CTCTTCCCCATCGGCTTTGT-3′
R 5′-CGGGGATGGTCATCTTCTCG-3′
*IGFBP6*: F 5′-GAACCGCAGAGACCAACAGA-3′
R 5′-GCAGCACTGAGTCCAGATGT-3′
*TGM2*: F 5′-CAGTCTCACCTTCAGTGTCGT-3′
R 5′-AAAGCTGGATCCCTGGTAGC-3′

### Transfections and generation of stable cell lines

Transient DNA transfections were performed using Torpedo DNA Transfection Reagent (Ibidi) according to manufacturer's instructions. The following plasmids were used: pCMVflag-Plk4 as a gift from Ingrid Hoffmann (DKFZ, Heidelberg, Germany); pSG5-hERβ kindly provided by P Chambon and H Gronemeyer (Institute for Genetics, and Cell and Molecular Biology, Strasbourg, France); pcDNA3-H2B-GFP (91); and pRP-hGPER1-siRES (VectorBuilder). All siRNA transfections were carried out using 60 pmol siRNA targeting *GPER1* or *SCRAMBLED* (Qiagen or Eurofins) using the INTERFERin siRNA transfection reagent (Polyplus Transfection) according to the manufacturer's protocol. GPER1 rescue experiments were performed by sequential transfection of cells with siRNAs, followed by the ectopic expression of the siRNA-resistant version of *GPER1* 4 h after siRNA transfection, and (xeno)estrogen treatment with or without 2 mM thymidine after an additional 4.5 h. To generate HCT116 cells with low *GPER1* expression, stable transfections of shRNA-expressing plasmids were performed using Metafectene (Biontex) according to the manufacturer's protocol using 2 µg of pRetro-Super plasmids targeting *GPER1*. Transfected cells were diluted (1:100–1:1,000), and single-cell clones were isolated and expanded on selection in a medium containing 1 µg/ml puromycin. The *GPER1* repression of selected cell clones was verified using qRT-PCR. For the long-term treatment of HCT116 cells with low *GPER1*

expression, puromycin-resistant single-cell clones stably expressing pRetro-Super empty vector or *GPER1* shRNA were long-term–cultivated in a medium containing puromycin (1 μg/ml) plus 10 nM E2, BPA, or DES.

## FACS analyses and determination of the mitotic index

Cells were fixed in 70% ethanol overnight at 4°C and resuspended in propidium iodide (5 μg/ml) and RNaseA (1 μg/ml) in PBS. FACS analyses were performed on a BD FACSCanto II (BD Biosciences), and 10,000 events were counted. Data analyses were performed using the BD FACSDiva software (BD Biosciences) and adapted with FlowJoTM 10.7.1. Representative examples of cell cycle profiles that are shown in the figures were repeated at least three times. The mitotic index was determined by staining of fixed cells with anti-MPM2 (1:1,600; Millipore) and secondary antibodies conjugated to Alexa Fluor 488 (1:2,000; Molecular Probes) as described previously (91).

## Proliferation assays

For quantitative proliferation assays, 5 × 10⁴ HCT116 or 1 × 10⁵ CCD 841 CoN cells were seeded in six-well plates in a stripped FCS medium. Cells were washed once they had attached to the surface and pretreated with DMSO or 100 nM G15 for 30 min after additional treatment with DMSO, 10 nM E2, BPA, DES, or 100 nM G1 for 7 d. The medium was changed every 2–3 d (with ligands). From day 2 on, cells were trypsinized and counted manually using a hemacytometer (purchased from Fein-OPTIK). To exclude dead cells in the counting, trypan blue was added in a ratio of 1:2.

## Protein extract preparation and Western blotting

To prepare whole-cell extracts, cells were washed once with PBS and detached from the plate with trypsin/EDTA solution (0.05%/0.02%[wt/vol]; PAN-Biotech), and the cell pellet was collected by centrifugation at 130*g* for 5 min at RT. Cells were lysed for 20 min in ice-cold lysis buffer (50 mM Tris–HCl [pH 7.4], 150 mM NaCl, 5 mM EDTA, 5 mM EGTA, 1% [vol/vol] Nonidet P-40, 0.1% [wt/vol] SDS, and 0.1% sodium desoxycholate) supplemented with protease inhibitor and phosphatase inhibitor mixtures (MilliporeSigma). For the generation of GPER1 lysates, extracts were also sonicated (Hielscher Ultrasonics GmbH) 20 times for 0.5 ms at 80% amplitude. Lysates were centrifuged at 20,000*g* at 4°C for 20 min. The supernatant was transferred into a new tube, and protein concentration was measured using the DC Protein Assay Reagents Package (Bio-Rad) according to the manufacturer's instruction. The concentration was determined photometrically at 750 nm using a TECAN Infinite M200 plate reader. 50 μg of protein was boiled with 5x sample buffer (250 mM Tris–HCl [pH 6.8], 50% [vol/vol] glycerol, 15% [wt/vol] SDS, 25% [vol/vol] ß-mercaptoethanol, and 0.25% [wt/vol] bromophenol blue) and loaded onto 10% SDS–PAGE with Tris running buffer and transferred to a nitrocellulose or PVDF membrane (GE Healthcare) using semidry or tank-blot procedures. Membranes were blocked with 6% nonfat powdered milk in 1% Tween in Tris-buffered saline. For Western blot experiments, the following antibodies and dilutions were used: anti-flag (F3165, 1:700; MilliporeSigma); anti-GPER1 (1:500, ab154069; Abcam); anti-ERα (1:500, sc-8002; Santa Cruz); anti-

ERβ (1:500, sc-53494; Santa Cruz); anti-α-tubulin (1:2,000, sc-23948; Santa Cruz); and anti-actin (clone AC-15, 1:60,000; Sigma-Aldrich). Secondary anti-mouse or anti-rabbit antibodies conjugated to HRP were used at 1:10,000 (111-035-146, 111-035-144; Jackson Immuno-Research). Proteins were detected by enhanced chemoluminescence. The quantification of Western blot bands was performed using the ImageJ software.

## Detection of centrosome amplification

For the quantification of amplified centrosomes, asynchronously growing cells were treated with estrogenic substances or GPER1 agonists (G-1, Tam, or ICI) for 48, 96, or 144 h, or for 30 generations where indicated. To block GPER1 activity, cells were pre-incubated for 30 min with G15 or G36 before additional (xeno)estrogen or control treatment. For *GPER1* repression, cells were first transfected with control or *GPER1*-specific siRNAs (Q2, CCD 841 CoN, #1 HCT116 and HCT-15) and subsequently treated with estrogens for 48 h. As a positive control, HCT116 cells were transiently transfected with pCMVflag-*PLK4* (kindly provided by I Hoffmann, DKFZ, Heidelberg, Germany). To visualize γ-tubulin, interphase cells were fixed with 2% PFA in PBS for 5 min at RT, followed by extraction with methanol at −20°C for 5 min. Cells were washed once with PBS and blocked with 5% FCS for 20 min. Subsequently, the cells were stained for γ-tubulin (1:650, GTU-88; MilliporeSigma). For the visualization of centrioles, cells were either fixed with methanol at −20°C for 6 min, followed by 3-min incubation with 0.3% Triton X-100 at RT (Cep135), or pre-incubated at 4°C for 40 min, followed by incubation with 1% PFA for 10 min at RT (CP110). Cells were washed twice with PBS supplemented with 0.05% Tween, and blocked with 1% BSA fraction V, protease free (Roth) for 30 min at RT. For permeabilization, cells were treated with 0.5% Triton X-100 for 40 s at RT, followed by treatment with methanol at 20°C for 20 min. Subsequently, the cells were stained for Cep135 (1:300, ab75005; Abcam) or CP110 (1:100, ab243696; Abcam) to detect the centrioles, for α-tubulin (1:650, C0415; Santa Cruz) to visualize microtubules, and with Hoechst (Hoechst 33342, 1:15,000, H1399; Invitrogen) to identify nuclei. Secondary antibodies conjugated to Alexa Fluor 488/555 (1:1,000, A-11029, A-11034, A-21424, A-21428; Life Technologies) were used. The amount of interphase cells with more than two centrosomes was quantified, that is, more than two γ-tubulin signals or more than two γ-tubulin/Cep135, and γ-tubulin/CP110 signals (39, 40) localized at centrosomes, respectively.

## Detection of centriole overduplication and premature disengagement/loss of centriole segregation

To detect "templated centriole overduplication" or centriole segregation defects during the onset of the M phase (likely indicating premature centriole disengagement), cells were blocked twice with 2 mM thymidine (Sigma-Aldrich). For templated centriole overduplication, cells were washed with fresh culture medium for 5 min and subsequently released into a medium supplemented with DMSO, or 10 nM E2, BPA, or DES for an additional 130 min. The S-phase release was verified by FACS analyses using propidium iodide as described in the Materials and Methods section "Determination of lagging chromosomes." Cells were fixed as described

previously (47). In short, cells were pre-extracted for 40 s in 0.5% Triton X-100 in BRB80 (80 mM K-Pipes, 1 mM MgCl$_2$, and 1 mM EGTA), washed once with PBS, and fixed with methanol for 7 min at −20°C. Cells were blocked with 5% BSA/PBS (sterile-filtered; Carl Roth) for 30 min. To visualize centrosomes and centrioles, cells were stained with antibodies against γ-tubulin (1:650, T3559; Merck Sigma) and Sas-6 (1:300, 91.390.21, sc-81431; Santa Cruz) diluted in 1% BSA/PBS and 0.2% Triton X-100 for 2 h. Subsequently, the cells were stained with Hoechst (Hoechst 33342, 1:15,000, H1399; Invitrogen) to identify nuclei. Secondary antibodies conjugated to Alexa Fluor 488/555 (1:1,000, A-11029, A-21428; Life Technologies) were used. Images were recorded with the ×63, 1.42 oil immersion objective with 40 z-stacks and a z-optical spacing of 0.28 $\mu$m under a Zeiss AxioObserver Z1 microscope (Zeiss). Images were acquired under constant exposure in each channel for all of the cells. Images were processed with ImageJ and shown as maximum-intensity projections. Analysis of cells for templated centriole overduplication was performed with CellProfiler (version 2.1.1). The mean centrosome intensity of HsSas-6 was quantified by drawing a region over the centrosomes, automatically identified as primary object by the software (see Fig 2F, inner red circle), and by measuring the mean integrated intensity values of γ-tubulin (MI$_\gamma$) and hsSas-6 (MI$_S$). To remove microscope noise, the background was measured on areas that expand the centrosome region by six pixels (see Fig 2F, outer red circle) after subtraction from MI$_\gamma$ values. The mean intensities of HsSas6 were calculated from the ratio of background-corrected HsSas-6 and γ-tubulin for each cell. The geometric mean was calculated from 212–241 cells of three independent experiments (10 nM [xeno]estrogens) and 89–178 cells of one experiment from five different coverslips (1 $\mu$M [xeno]estrogens).

To detect centriole segregation defects as an indication of premature disengagement, cells were released from the double thymidine block into a fresh culture medium for 4.5 h (to pass S phase). Subsequently, cells were treated as before for 4 h (i.e., from G2 to metaphase). Cells were fixed, stained, and imaged (up to 30 z-stacks) as described above. Hoechst 33342 staining was used to identify metaphase chromosomes. Images were processed with the ZEN 3.1 blue edition software (Carl Zeiss Microscopy GmbH) and shown as maximum-intensity projections. Metaphase cells were examined for the presence or absence of Sas-6 signals positive for PCM markers (i.e., γ-tubulin), as the latter could indicate premature centriole disengagement.

### Determination of lagging chromosomes

To detect lagging chromosomes in the anaphase of mitosis, cells were either treated with 10 nM E2, BPA, DES, or 5 nM nocodazole (9) alone or, where indicated, exposed to 100 nM G15 for 30 min (CCD 841 CoN, HCT-15; RKO) or transfected with control or *GPER1* siRNAs for 24 h (#1, HCT116; #1 or Q2, CCD 841 CoN) followed by (xeno)estrogen treatment. For synchronization in anaphase, HCT116, HCT-15, and RKO cells were blocked twice with 2 mM thymidine (Sigma-Aldrich) and subsequently released into fresh medium for 8–9 h as described previously (91). CCD 841 CoN cells were treated and grown asynchronously for 48 h. Cells were fixed with 2% PFA in PBS for 5 min at RT followed by extraction with methanol at −20°C for 5 min. To visualize kinetochores, cells were stained with antibodies

against centromeres (CREST, HCT-0100, 1:200; Europa Bioproducts). Chromosomes were stained with Hoechst 33342 (MilliporeSigma). A secondary antibody conjugated to Alexa Fluor 555 (1:1,000, A-21433; Life Technologies) was used. Cells were analyzed by immunofluorescence microscopy, and chromosomes that were CREST-positive and clearly separated from two pole-oriented chromosome masses were counted as "lagging chromosomes." Cell cycle profiles and the mitotic index were determined of ethanol-fixed cells, treated with RNaseA (1 $\mu$g/ml) for 30 min followed by immunostaining with anti-MPM2 antibodies (1:1,600, 05-368; MilliporeSigma) and treatment with propidium iodide (5 $\mu$g/ml) for 30 min. FACS analyses were performed using the BD FACSDiva software (Becton Dickinson).

### Karyotype analyses and Cep-FISH

HCT116 single-cell clones or CCD841CoN and HCT-15 cell populations were treated with DMSO, E2, BPA, DES, or Noc for 30 generations. To arrest cells in mitosis, cells were treated with 300 nM Noc or 2 $\mu$M dimethylenastron (MilliporeSigma) for 4 h (HCT116, HCT-15) or 2 $\mu$M dimethylenastron for 7 h (CCD 841 CoN). Cells were harvested and hypotonically swollen in 40% RPMI 1640 for 15–20 min at RT. Cells were fixed in Carnoy's fixative solution (75% methanol/25% acetic acid) and subsequently dropped onto cooled glass slides after drying at RT. Chromosomes were stained with 5% Giemsa solution (MilliporeSigma) for 10 min, rinsed with water, air-dried, and mounted with Euparal (Roth). The proportion of cells with chromosome numbers deviating from the modal number was determined. Chromosome number variability was also determined by interphase FISH. FISH was performed using α-satellite probes specific for chromosomes 2 and 8 (CytoCell) according to the manufacturer's protocol. Images were acquired as 0.24-$\mu$m optical sections with the ×63 1.4 NA objective, and chromosome signals in 100 nuclei were determined.

### Immunofluorescence microscopy

Microscopy of fixed cells was performed on a Zeiss AxioObserver Z1 microscope (Zeiss) equipped with an Apotome 2.0 module, a heated chamber, and an AxioCam MRm camera (Zeiss). Images were recorded with the ×63, 1.42 oil immersion objective and a z-optical spacing of 0.24 $\mu$m, processed with the ZEN 3.1 blue edition software (Carl Zeiss Microscopy GmbH), and shown as maximum-intensity projections. Image sections had a size of 25 × 25 or 35 × 35 $\mu$m. For the visualization of live chromosomes, HCT116 cells transfected with 10 $\mu$g of pcDNA3-H2B-GFP (91) were seeded in 12-well plates and treated with DMSO, 10 nM of each (xeno)estrogen, or 5 nM nocodazole once cells had attached to the cell culture surface. After 24 h, cells were seeded in 35-mm glass-bottom dishes (Ibidi) with four compartments in a stripped FCS medium and cultured overnight. A fresh stripped FCS medium (with ligands) was added before cells were transferred into the prewarmed microscope chamber. Images were recorded with the ×40, 1.42 objective, and a z-optical spacing of 2 $\mu$m was recorded every 2 min for 10 h. 10 frames for each condition were determined. The time-point of NEB was defined as the first frame showing the loss of smooth appearance of chromatin, and anaphase was the first frame when chromosomes move toward the cell poles. Mitotic delay was defined as the median time from NEB to anaphase

equal to or greater than 1.5-fold of the median time observed in DMSO-treated control cells (22 min). Box-and-whisker plots with 5–95 percentile were calculated from image sequences from at least 400 recorded cells (100 for each experiment) using the PRISM 8.2.0 software.

### Statistical analysis

All quantifications of CA and lagging chromosomes are based on three to six independent experiments (if not stated otherwise), in which 300–1,200 interphase cells (CA) or 500–600 anaphases (lagging chromosomes) were evaluated, and mean values with standard error of the mean (SEM) or median with 95% confidence interval (qRT-PCR) or median with box and whiskers with 5–95 percentile were calculated. All karyotype analyses are based on the quantification of individual chromosome numbers from 25 to 50 metaphase spreads. No data were excluded from the analysis. Investigators, who obtained these data, were not blinded to substance treatment. Statistical analyses of most experiments were performed with R.4.0.2. *P*-values for centriole overduplication (Mann–Whitney's test) and *P*-values for FACS analyses (ordinary one-way ANOVA) were calculated using GraphPad Prism 6. Data from most experiments are (i) count data dealing with (ii) fixed effects (substance treatments) and (iii) random effects (biological replicates). The generalized linear mixed model (*glmm*) considers all three conditions (i–iii). In detail, *glmm* was applied to test the effect of substance treatments on the probability $\pi$ of the formation of cells with CA or lagging chromosomes:

$$\pi = \frac{\exp \eta}{1 + \exp \eta},$$

$$\eta = X\beta + Zb + \varepsilon.$$

Here, $\beta$ denotes a coefficient vector of the substance treatment, X is the corresponding fixed-effect matrix, $b$ is the coefficient vector of the biological replicates with random-effect matrix Z, and $\varepsilon$ denotes the technical error. Here, $b$ and $\varepsilon$ are regarded as normally distributed with zero mean and standard deviations $\sigma_b$ and $\sigma_\varepsilon$.

For the estimation of the *glmm*, the function *glmmTMB* (92, 93) from the R package glmmTMB was used. The null hypothesis $\beta_0 \geq \beta_i$, that is, the hypothesis that the substance *i* has the same or smaller impact on $\pi$ as the control, was tested by Wald's z-statistics and its corresponding *P*-values. To test the null hypothesis that there is no difference in $\pi$ resulting from n different substance treatments ($\beta_0 = \beta_1 = .. = \beta_n$), two different glmms were compared with the R function *ANOVA*. The first model regards the common mean only ($\boldsymbol{\eta} = \mu + Z\boldsymbol{b} + \varepsilon$) as a fixed effect. The second model regards the substance treatment as a fixed effect ($\boldsymbol{\eta} = X\boldsymbol{\beta} + Z\boldsymbol{b} + \varepsilon$). The biological replicates were regarded in both models as random effects.

If for the control or one treatment the number of cells with CA was zero for all biological replicates, *P*-values were estimated by the following *bootstrap* procedure: for the control and each treatment, *i* probabilities of CA were estimated by

$\pi_{ij}$ = (number of cells with centrosome amplification)/ (number of all observed cells)($j = 1, .., n_b$).

From this, $n_b \times 100{,}000$ probabilities were drawn with a replacement for each treatment and the control. With these probabilities, the number of cells with centrosomes ($n_{ij}$) was randomly chosen from the corresponding binomial distribution. The number of cases, where the sum over the $n_b$ numbers of cells with CA for a treatment was equal or smaller than the corresponding number for the control, was counted. This number divided by 100,000 is then the *P*-value.

Standard deviations and median values of CA with substance treatment *i* were estimated by the following *bootstrap* procedure: for each of the $n_b$ biological replicates, this probability was estimated by

$\pi_{ij}$ = (number of cells with centrosome amplification)/ (number of all observed cells)($j = 1, .., n_b$).

$n_b$ probabilities were 1,000 times randomly drawn with replacement from the above-described set of $n_b$ estimated probabilities. For each of the chosen estimates, the number of cells with CA was 10 times chosen randomly according to the binomial distribution with probability $\pi_{ij}$ and size = n (number of all observed cells). Therefore, we have obtained 10,000 simulated estimates of $\pi_i$. The median and SD are calculated from this set.

Fig S4B: The effect of substance treatment on the proportion of cells with a time from NEB to anaphase more than 1.5 that of the median time observed in the DMSO control is tested in the same way as described with the R function *glmmTMB*. Standard deviations and medians were estimated by the bootstrap procedure described above.

ns, not significant; *P < 0.05; **P < 0.01; ***P < 0.001; and ****P < 0.0001. *P*-values are given online.

## Data Availability

This study includes no data deposited in external repositories.

## Supplementary Information

## Acknowledgements

We thank Ingrid Hoffmann (DKFZ, Heidelberg, Germany), Pierre Chambon, and Hinrich Gronemeyer (Institute for Genetics, and Cell and Molecular Biology, Strasbourg, France) for plasmids; Beate Döring, Birgitta Slawik, Sarah Schmerbeck, and Maria Gabriel for technical assistance; Sebastian Dunst for sharing reagents; Norman Ertych for support with CellProfiler; and all colleagues from the BfR/Bf3R, especially Gilbert Schönfelder, Michael Oelgeschläger, Sebastian Dunst, Shu Liu, Norman Ertych, and Marta Barenys Espadaler, for scientific input and/or comments on the article. This work was supported by an intramural funding of the German Federal Institute for Risk Assessment (BfR) provided to M Bühler and A Stolz (SFP No.1322-733) and funded by the Deutsche Forschungsgemeinschaft (DFG, German Research Foundation)—Project No. 465732234; STO 1306/2-1 to A Stolz.

## Author Contributions

M Bühler: conceptualization, data curation, formal analysis, investigation, methodology, and writing—original draft, review, and editing.
J Fahrländer: data curation, formal analysis, investigation, methodology, and writing—review and editing.
A Sauter: conceptualization, data curation, formal analysis, investigation, methodology, and writing—review and editing.
M Becker: data curation, formal analysis, investigation, methodology, and writing—review and editing.
E Wistorf: data curation, formal analysis, investigation, methodology, and writing—review and editing.
M Steinfath: formal analysis, statistics, and writing—review and editing.
A Stolz: conceptualization, data curation, formal analysis, funding acquisition, methodology, project administration, and writing—original draft, review, and editing.

## Conflict of Interest Statement

The authors declare that they have no conflict of interest.

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
