## [Reviewer comments · Life Science Alliance]

Life Science Alliance

GPER1 links estrogens to centrosome amplification and chromosomal instability in human colon cells

Miriam Bühler, Jeanine Fahrländer, Alexander Sauter, Markus Becker, Elisa Wistorf, Matthias Steinfath, and Ailine Stolz
DOI: <https://doi.org/10.26508/lsa.202201499>

Corresponding author(s): Dr. Ailine Stolz (Federal Institute for Risk Assessment)

Review Timeline:

Submission Date:	2022-04-24
Editorial Decision:	2022-05-31
Revision Received:	2022-09-27
Editorial Decision:	2022-10-18
Revision Received:	2022-10-24
Accepted:	2022-10-25

Transaction Report:

May 31, 2022

Re: Life Science Alliance manuscript #LSA-2022-01499-T

Dr. Ailine Stolz
Federal Institute for Risk Assessment
Experimental Toxicology and ZEBET
Max-Dohrn-Str. 8-10
Berlin 10589
Germany

Dear Dr. Stolz,

Thank you for submitting your manuscript entitled "GPER1 links estrogens to centrosome amplification and chromosomal instability in human colon cells" to Life Science Alliance. The manuscript was assessed by expert reviewers, whose comments are appended to this letter. We invite you to submit a revised manuscript addressing the Reviewer comments.

Thank you for this interesting contribution to Life Science Alliance. We are looking forward to receiving your revised manuscript.

Sincerely,

B. MANUSCRIPT ORGANIZATION AND FORMATTING:

Reviewer #1 (Comments to the Authors (Required)):

Reviewer comments:

This is an interesting study by Buhler and colleagues of the link between estrogens, alternate estrogen receptor GPER1, centrosome amplification, erroneous mitosis and chromosomal instability in human colon cell lines. The authors combine treatment of various CRC-derived and normal colon epithelial cell lines with different xenoestrogen compounds to observe their effect on centrosome amplification, an important and well-established mechanism of high chromosomal instability in CRC. The authors observed that treatment with estrogenic substances increased the proportion of cells with multiple centrosomes and centrioles 2-4-fold, in all cell lines tested irrespective of their CRC-derived or normal colon epithelial origin. Interestingly, the authors noticed that the effect of xenoestrogens on centrosome amplification depends on activated GPER1 receptor in all cell lines, as its partial depletion or inhibition rescued centrosome amplification observed after treatment in each cell line. Importantly, activation of GPER1 through other pathways resulted also in 2-3-fold increase in number of cells with centrosome amplification. To observe direct effects of centrosome amplification on mitosis, the authors live-imaged HCT116 cells treated with estrogens and observed transient multipolar mitoses, and unaligned chromosomes, that led to induction of strong mitotic delays in subpopulation of treated cells. By analysis of synchronized and fixed anaphase cells after treatment with estrogenic compounds, the authors observed a 2-3-fold increase in the number of lagging chromosomes, a well-established consequence of centrosome amplification and transient multipolarity. Authors furthermore show that continuous treatment of cell cultures with estrogens caused 2-3-fold increase in the number of aneuploid cells and whole chromosome mis-segregations. Finally, the authors showed that even this long-term aneuploidy phenotype depends on GPER1 activation, as its partial depletion or inhibition resulted in the rescue of observed aneuploidy, while the rescue of aneuploidy was not observed after treatment of cells with low dose of nocodazole.

CRC biology has been studied in depth by in situ approaches based on cell lines, and some of the pathways for generating high CIN in CRC described in the manuscript are well established in the field, as the authors nicely present by themselves. Furthermore, the effect of estrogenic signaling on centrosome amplification and mitotic spindle was previously described in different contexts. However, despite the rich knowledge on the mechanisms connecting CA and CIN, I found the results of this study innovative, well presented, and surely worth publishing in the Life Science Alliance. First, the study will be of interest for researchers working on mechanisms of CIN, especially those interested in CIN-generating pathways in human colon cells, as well to researchers working on estrogenic signaling and its effects on tumorigenesis in general. The work nicely follows the tremendous work done on the role of centrosome amplification, transient multipolar spindles, and lagging chromosomes in generating chromosome mis-segregation and aneuploidy in the context of tumorigenesis. Finally, the presented study linked in an experimental fashion many different and well-accepted aspects of CRC biology, but introduced a few on its own, like a clear link to specific GPER1 receptor, that could be studied in more detail later.

The manuscript is well-written, and the results are very clean and clear for the most part except for a few instances emphasized in the comments below. Although I think that the manuscript presents interesting results that could allow for more detailed studies of the effect of estrogenic signaling and GPER1 on CRC, the authors would need to review the manuscript in a way that better illustrates certain points presented below. If authors would succeed in presenting the clearer picture of main findings of the paper, by introducing new measurements and more relevant Discussion, and where possible new experiments, my opinion is that the manuscript would be quite at the Life Science Alliance level.

Major comments:

1) Authors should discuss in much more detail observation that continuous treatment of cell culture with estrogenic compounds did not have any apparent effect on cell cycle progression and distribution of cell cycle stages, as it is not a simple phenotype that could be ignored. My view is that this is a very important point, as the authors observed an increased number of centrosomes (Fig. 1), prolonged mitosis (Fig. 3), increased aneuploidy, and whole-chromosome missegregations (Fig. 4) after treatment with xenoestrogens, and all of these are potent activators of cell cycle arrest. For example, PLK4 overexpression that causes centrosome amplification decreased cell proliferation via the p53-p21-dependent pathway in p53-proficient cells (Holland et al., 2012, Genes&Development), and a similar assay is used in this study. To my knowledge, the cell lines used in the presented study are all p53-proficient. Could it be that treated cells naturally tend to lose extra centrosomes after extensive passaging, as observed by Ganem et al., 2009, Nature for RPE1 and BJ cells. Furthermore, it could be that daughter cells of a

mother with prolonged mitosis, that are observed also in this study (Fig. 3), cannot divide further in p53 proficient cells irrespective of a cause of prolongation, as observed by Uetake&Sluder, 2013, Current Biology. This is certainly not favored as authors observed similar distribution of cell cycle stages upon continuous treatment with xenoestrogens. I guess my question is if the number of cells with more than two centrosomes change with passaging of cells in presence or absence of estrogens, and how? The authors should discuss possible mechanisms and implications or perform new experiments similar to those in Fig. 1 but with few time points at which they could follow the proportion of cells with CA.

2) In relation to the previous comment, the authors observed a low percentage of cells with centrosome amplification (max. 10%) even after prolonged treatment with inhibitors, and the authors do not describe the reason for this. This is followed by a similarly low percentage of cells with lagging chromosomes and increased mitotic durations, implying that cells that did not show an increased number of lagging chromosomes and increased mitotic durations are those that have normal number of centrosomes. An alternative explanation is that for some unknown reason certain cells do not respond to this treatment, but that is probably not the case, as only slightly higher percentage of CA cells is observed after overexpression of PLK4 (S1C). If time permits, the authors could synchronize the cells before treatment to enable synchronous passage of the population through the S phase, possibly on only one representative cell line. In this way, the authors could observe whether xenoestrogens act on all cells in S phase equally or whether there are other factors that are more essential to maintain the number of centrosomes. Also, authors should quantify the proportion of cells with transient multipolar H2B phenotype and unaligned chromosomes during pseudo-metaphase (see minor comment 4) in live-cell imaging experiments after treatment with xenoestrogens, now only represented as images on Fig. 3A. This could be a good indication of how effective treatment is and whether cells that showed strong delays in mitosis are those that have transient multipolar spindles that are connected with amplified centrosomes (Ganem et al., 2009, Nature). The authors should also indicate that the increased number of unaligned chromosomes could be a possible cause of the increase in the number of lagging chromosomes (see Vukušić&Tolić, 2022, Cells).

Minor comments:

- 1) In the Introduction, page 1, line 26 the authors state that lagging chromosomes randomly segregate to new daughter cells. This is only partially true as there are increasing number of papers that reported non-random inheritance of lagging chromosomes that is influenced by age of centrosome in daughter cells (Gasic et al., 2015, eLife), activity of Aurora B within spindle midzone (Sen et al., 2021, Developmental Cell; Orr et al., 2021, Current Biology), or number of microtubules attached to each sister and midzone-derived forces during anaphase B in general (Cimini et al., 2004, Current Biology). Authors should acknowledge the mentioned papers or change this sentence.
- 2) In a sentence on page 3, line 24 ('Cells with...'), the authors should cite the fundamental work of the David Pellman group linking centrosome amplification, transient multipolar spindles, and lagging chromosomes in human cells (Ganem et al., 2009, Nature).
- 3) Regarding the observation that CA increased mitotic duration, the authors should cite the fundamental work from the Conley Rieder lab, since it showed that an increase in the number of centrosomes increases the duration of mitosis in normal human cells irrespective of other changes (Yang et al., 2008, Nature Cell Biology).
- 4) On page 11, line 43 the authors use the term 'prometaphase-like state' to refer to metaphase cells with unaligned chromosomes. I would advise the authors to use the term 'pseudo-metaphase', a widely accepted term for such mitotic cells, as defined by Weaver et al., 2003, J Cell Biol.
- 5) On page 11, line 38, the authors state that cells treated with xenoestrogens 'progressed through mitosis and segregated their chromosomes, albeit with reduced fidelity'. The term reduced fidelity should be much more precise. I would suggest that the authors use the term 'unaligned chromosomes,' as they did in the figure caption, which are only transiently present in treated cells during pseudo-metaphase in this study, as such chromosomes could drastically reduce the fidelity of mitosis.
- 6) I would encourage the authors to include additional examples of live-imaged mitotic cells in Fig. S3, similar to those presented on Fig. 3A. I would also encourage authors to enlarge the live-cell H2B figures, as currently they are in a too small format.
- 7) I did not understand why authors did not quantify the proportion of cells with laggards/chromosome bridges from live-cell movies of treated, and untreated cells. If there is some specific reason for this, please include it in the Results section, otherwise authors should quantify the proportion of such chromosomes in a population of imaged cells.
- 8) On page 12, line 1 the authors should cite the work from Daniel Gerlich lab that showed how increased mitotic duration observed after treatment of cells with low concentrations of nocodazole scales with the number of unaligned chromosomes (Dick&Gerlich, 2013, Nature Cell Biology).
- 9) In the caption of Figure 3C, include the exact concentration of nocodazole used, as it is critical to the observed degree of mitotic delay after this treatment (see previous comment).
- 10) It is unclear to me from the caption of Fig. 3A and the M&M section "Immunofluorescence microscopy" whether the cells were live imaged in the presence of xenoestrogens. Please rephrase the sentences to make this clearer or introduce a scheme of experimental design in Fig. 3A or Fig. S3.
- 11) Related to the major comments presented above, the authors should discuss why the number of aneuploid cells observed in long-term treatments that span 30 generations (Fig. 4) is much larger than the proportion of cells with CA (Fig. 1), and cells with lagging chromosomes (Fig. 3) after short treatments. Related to that point, what would authors expect if cells were treated with estrogens for a short time period and then left for 30 generation? The authors should discuss implications and possible mechanisms of coping with centrosome amplification in this context.
- 12) Cite appropriate papers for sentence on page 19, line 3 ('Our results hypothesize...').
- 13) The authors should further discuss the implication of their observation that CRC-derived and normal colon epithelial cells behave similarly after xenoestrogen treatment in this study, except the fact that percentages of cells with CA and lagging

chromosomes are lower in normal colon cells. Interestingly, the amount of aneuploidy observed in the population is similar after long-term treatment with xenoestrogen in HCT116 and CCD 841 CoN cells.

14) Label treatments in the left part of Figure S3D.

15) The 'Vinca alkaloid' is mentioned on page 12, line 1, but there are no data for it.

16) On page 12, line 7, thymidine is misspelled as thymine.

17) Figure S1D could be placed in a main figure, as it implicated PLK4 as a bottom target of estrogen-GPER1 signaling, which is essential for centrosome amplification.

18) Graphical cover should not be depicted as a circle as it is not a feedback loop since chromosomal instability does not lead to GPER1 activation.

19) The authors should calculate error bars and statistical significance of results presented on Figures 4C, 5C, 5D, S4B, and S5D.

Reviewer #2 (Comments to the Authors (Required)):

Buehler, Stolz and colleagues here explore the impact of (xeno-)estrogens on centrosomes and chromosome stability in colon cancer cells and colon epithelial control cells. They demonstrate the induction of centrosome abnormalities by GPER1-activating estrogen treatment that are accompanied by mitotic problems and chromosome instability. These data are of potential relevance in how estrogens may contribute to cancer development in the colon. The findings are novel and should interest researchers in a several fields.

In general, the experiments in the paper are thoroughly done and well described; the key conclusions of the study are supported by the data presented.

I suggest some additional control experiments and clarifications that should be included to strengthen the findings presented here:

1. A frequency distribution of the centriole numbers should be provided, based on counts with CEP135 or CP110. Without any detectable impact on the cell cycle distribution of the treated populations, it should be tested whether the treatment outcomes are due to loss of centriole cohesion or CA. Related to this point, do increasing doses of estrogens cause increasing centriole numbers?
2. A rescue control for the siRNA experiments in Figures 2 and 5 should be provided.
3. The data in Fig. S3D are particularly important and should be included in the main manuscript. These data exclude the possibility that an estrogen-regulated cell cycle arrest causes the centrosome phenotype. The information in this Figure should be improved: the 4 panels in the FACS plot should be labelled for their respective treatments and the description of the cell cycle phases should reflect those in the Figure itself -here are 4 categories of cell in the graph, but only 3 mentioned in the legend; and it seems that sub-G1 cells are not included in the graph (?). It should be stated how many times this experiment was repeated and relevant statistics included in the Figure.

Cross comments on referee reports:

I agree with the other 2 reviewers' point on the need for a more extended timecourse of centrosome numbers over treatment time. This (and the related interpretation of the results, of course) would provide important new information. I suggest that the authors count centrioles during this timecourse; this point was also raised by referee 3.

It may be challenging to synchronise the cells (as suggested by referee 1) without impacting on centriole numbers. The authors should consider carefully how to approach this suggestion.

Reviewer #3 (Comments to the Authors (Required)):

In this manuscript the authors demonstrate that in normal and cancerous colonic epithelial cells estrogen-dependent stimulation of the G protein coupled estrogen receptor GPER1 is responsible for supernumerary centrosomes, aneuploidy and an increased variability in mitotic duration. Intriguingly, estrogens via GPER1 seem to trigger chromosomal instability at levels comparable to that caused by low doses of nocodazole. Overall the data on the role GPER1 in karyotype stability is convincing, however, additional experimental data is needed to show if this is indeed due to centrosome amplification.

1. The centrosome amplification phenotype shown in Fig 1 and Fig S1 is interesting, although it is fairly infrequent in the population (remains under 10% despite 48 hr treatment). My concern is that all the images (throughout the manuscript) depict precisely three PCM-containing centrosomes (which means that these centrioles would have been produced in the preceding cell cycle or before). Normally, overduplication involves either de novo assembly of centrioles, or parental centrioles templating assembly of more than 1 centriole each. It is difficult to imagine that such pathways yield 3 centrosomes. In my opinion the phenotype the authors see could be due to unequal segregation of centrosomes between daughter cells with

one cell inheriting three and the other a single centriole. Such a phenotype arises if a parental centriole and its procentriole split apart (i.e. disengage prematurely) during mitosis and the four centrioles segregate unevenly. These split centrosomes may even promote formation of additional spindle poles, so the transient multipolarity the authors allude to would fit this model. Moreover, a mitotic delay can trigger centrosome splitting and abnormal centriole/centrosome segregation, and thus the centrosome phenotype may actually be secondary to a mitotic defect. To distinguish between these possibilities the authors should score centrosome numbers more precisely (i.e. whether cells contain 0, 1, 2, 3, 4 or more centrosomes). If it is confirmed that many cells have more than 3 centrosomes, then trying a procentriole marker like Sas6 will help establish if parental centrioles indeed assemble more than one centriole each and if de novo assembly of procentrioles occur.

2. The multipolarity phenotype should be shown with centrosome and microtubule markers as well to establish if extra centrosomes are driving this phenotype.

3. It would be crucial to show centrosome number evolution in the clones that were used for karyotype analysis following 30 generations.

4. It is perplexing that all the treatments including nocodazole give rise to a nearly identical level of aneuploidy (30-40% of cells) both in HCT116 and normal colon cells. Could it be that this is the limit at which the cells can still proliferate?

Bühler et al., manuscript LSA-2022-01499-T

GPER1 links estrogens to centrosome amplification and chromosomal instability in human colon cells

We cordially thank all reviewers for an excellent review of our manuscript. We address the reviewer's comments point-by-point below along with a specification of all changes included in the revised version (blue lettering). Unfortunately, the Corona pandemic progressed soon after we received the referee comments, which led to a delay in the laboratory work. Now, we are pleased to resubmit a significantly improved manuscript, which now includes 6 Figures with a total of 38 new panels:

13 new panels in the main figures (including 1 completely new figure)

25 new panels in the supplemental figures (including 1 completely new supplementary figure)

A summary of the major alterations are as follows:

- New panels (C and D) are included to **Fig S1** to follow the proportion of cells with CA over time (referees #1 & #2). **Previous Fig S1D** is now transferred to **Fig 2E**
- Current **Fig 2** includes nine new panels illustrating the presence of mainly three PCM-containing centrosomes (A-D) and centriole overduplication by means of increased Sas-6 fluorescence intensity at S-phase centrosomes (F-I), as an underlying mechanism of (xeno)estrogen-triggered CA (referee #2 and #3).
- Novel **Fig S2** illustrates a frequency distribution of the centrosome and centriole numbers based on counts with γ -tubulin, Cep135 and Cep110 (A-C) (referee #2). Panels D-G are included to examine whether premature disengagement of centrioles during early mitosis could be a possible mechanism of (xeno)estrogen-induced CA, as suggested by referees #2 & #3. We more likely ruled out this mechanism because daughter centrioles (visualized by Sas-6) segregated evenly to the cell poles (i.e., each cell pole contained at least one Sas-6 signal). As suggested by referee #2, we included a further panel (H-I), which examines whether increasing concentrations of (xeno)estrogens do increase centriole numbers, here addressed by measurement of increases Sas-6 intensities at centrosomes upon 1 μ M of estrogenic substances.
- **Previous version of Fig 2** is identical to current **Fig 3**
- **Previous Fig S2** is now shifted to **Fig S3**. The new panel **F** displays relative mRNA expression level of GPER1 for every condition in the GPER1 rescue experiment (referee #2). The two new panel **G** and **H** illustrate the GPER1 rescue experiment where the amount of cells with centrosome amplification was determined (referee #2). Data shown in **previous Fig S3D** are now included in **Fig 4**, panel G (referee #2). Treatments are now labeled (left part of panel G) and description of the cell cycle phases now reflect the Figure itself with 5 categories of cells in the graph and legend (referees #1 & #2). Data in the **previous Figure S3D** (right part) showed one representative experiment. We now

included data of the three different experiments in **Fig 4G** (right panel) with relevant statistics (referee #2).

- **Previous Fig 3** is now shifted to **Fig 4**. The multipolarity phenotype was quantified in synchronized cells labeled with centrosome and microtubule markers (new panel **C**) to show that extra centrosomes drive this phenotype in response to (xeno)estrogen-treatment (referee #1). We further include measurements of pseudo-metaphases (new panel **D**) to complement new data shown in Fig S4E.
- **Previous Fig 3C** is now shifted to **Fig S4B**.
- **Fig S4A** displays the experimental setup of H2B-GFP imaged cells (referee #1). New panel **S4C** includes additional examples of live-imaged mitotic cells after treatment with E2, BPA or nocodazole to complement images shown in current Fig 4A (referee #1). In addition, live-cell H2B-GFP figures are enlarged (referee #1). Three new panels (**D-F**) display quantification of the proportion of cells with multipolar mitoses, pseudo-metaphases and chromosome bridges from live-cell movies of treated and untreated cells (referee #1).
- **Previous Fig 4** is transferred to **Fig 5**. Error bars are now included in panel **C**, identical with previous Fig 4C (referee #1). Note that we are not able to give p-values as data are derived from 1 biological replicate. Two new panel (**E-F**) now display centrosome number evolution in stable HCT116 cell clones and CCD 841 CoN cells that were used for karyotype analyses following 30 generations of permanent treatment (referee #3).
- **Previous Fig S4** is transferred to **Fig S5**. Error bars are now included in new panel **C**, identical with previous Fig S4B (referee #1). Note that we are not able to give p-values as data are derived from 1 biological replicate. The new panels **D, E, and G** illustrate the frequency distribution of centrosome numbers of long term treated HCT116, CCDs or HCT-15 cells used for karyotype analyses. New panel **F** displays centrosome number evolution in HCT-15 cells that were used for karyotype analyses following 30 generations (referee #3).
- **Previous Fig 5** is shifted to **Fig 6**. Error bars are now included in panels **C** and **D**, identical with **previous Figs 5C** and **D** (referee #1). Note that we are not able to give p-values as data are derived from 1 biological replicate.
- **Previous Fig S5** is shifted to **Fig S6**. The two new panel **A** and **B** illustrate the GPER1 rescue experiment where the amount of cells with lagging chromosomes was determined (referee #2). Error bars are now included in panel **F**, identical with **previous Fig S5D** (referee #1). Note that we are not able to give p-values as data are derived from 1 biological replicate.

Reviewer #1 (Comments to the Authors (Required)):

Reviewer comments:

This is an interesting study by Buhler and colleagues of the link between estrogens, alternate estrogen receptor GPER1, centrosome amplification, erroneous mitosis and chromosomal instability in human colon cell lines. The authors combine treatment of various CRC-derived and normal colon epithelial cell lines with different xenoestrogen compounds to observe their effect on centrosome amplification, an important and well-established mechanism of high chromosomal instability in CRC. The authors observed that treatment with estrogenic substances increased the proportion of cells with multiple centrosomes and centrioles 2-4-

fold, in all cell lines tested irrespective of their CRC-derived or normal colon epithelial origin. Interestingly, the authors noticed that the effect of xenoestrogens on centrosome amplification depends on activated GPER1 receptor in all cell lines, as its partial depletion or inhibition rescued centrosome amplification observed after treatment in each cell line. Importantly, activation of GPER1 through other pathways resulted also in 2-3-fold increase in number of cells with centrosome amplification. To observe direct effects of centrosome amplification on mitosis, the authors live-imaged HCT116 cells treated with estrogens and observed transient multipolar mitoses, and unaligned chromosomes, that led to induction of strong mitotic delays in subpopulation of treated cells. By analysis of synchronized and fixed anaphase cells after treatment with estrogenic compounds, the authors observed a 2-3-fold increase in the number of lagging chromosomes, a well-established consequence of centrosome amplification and transient multipolarity. Authors furthermore show that continuous treatment of cell cultures with estrogens caused 2-3-fold increase in the number of aneuploid cells and whole chromosome mis-segregations. Finally, the authors showed that even this long-term aneuploidy phenotype depends on GPER1 activation, as its partial depletion or inhibition resulted in the rescue of observed aneuploidy, while the rescue of aneuploidy was not observed after treatment of cells with low dose of nocodazole.

CRC biology has been studied in depth by in situ approaches based on cell lines, and some of the pathways for generating high CIN in CRC described in the manuscript are well established in the field, as the authors nicely present by themselves. Furthermore, the effect of estrogenic signaling on centrosome amplification and mitotic spindle was previously described in different contexts. However, despite the rich knowledge on the mechanisms connecting CA and CIN, I found the results of this study innovative, well presented, and surely worth publishing in the Life Science Alliance. First, the study will be of interest for researchers working on mechanisms of CIN, especially those interested in CIN-generating pathways in human colon cells, as well to researchers working on estrogenic signaling and its effects on tumorigenesis in general. The work nicely follows the tremendous work done on the role of centrosome amplification, transient multipolar spindles, and lagging chromosomes in generating chromosome mis-segregation and aneuploidy in the context of tumorigenesis. Finally, the presented study linked in an experimental fashion many different and well-accepted aspects of CRC biology, but introduced a few on its own, like a clear link to specific GPER1 receptor, that could be studied in more detail later.

The manuscript is well-written, and the results are very clean and clear for the most part except for a few instances emphasized in the comments below. Although I think that the manuscript presents interesting results that could allow for more detailed studies of the effect of estrogenic signaling and GPER1 on CRC, the authors would need to review the manuscript in a way that better illustrates certain points presented below. If authors would succeed in presenting the clearer picture of main findings of the paper, by introducing new measurements and more relevant Discussion, and where possible new experiments, my opinion is that the manuscript would be quite at the Life Science Alliance level.

We appreciate the positive attitude and suggestions expressed by the referee.

Major comments:

1) Authors should discuss in much more detail observation that continuous treatment of cell culture with estrogenic compounds did not have any apparent effect on cell cycle progression and distribution of cell cycle stages, as it is not a simple phenotype that could be ignored. My view is that this is a very important point, as the authors observed an increased number of centrosomes (Fig. 1), prolonged mitosis (Fig. 3), increased aneuploidy, and whole-chromosome missegregations (Fig. 4) after treatment with xenoestrogens, and all of these are potent activators of cell cycle arrest. For example, PLK4 overexpression that causes centrosome amplification decreased cell proliferation via the p53-p21-dependent pathway in p53-proficient cells (Holland et al., 2012, Genes&Development), and a similar assay is used in this study. To my knowledge, the cell lines used in the presented study are all p53-proficient. Could it be that treated cells naturally tend to lose extra centrosomes after extensive passaging, as observed by Ganem et al., 2009, Nature for RPE1 and BJ cells. Furthermore, it could be that daughter cells of a mother with prolonged mitosis, that are observed also in this study (Fig. 3), cannot divide further in p53 proficient cells irrespective of a cause of prolongation, as observed by Uetake&Sluder, 2013, Current Biology. This is certainly not favored as authors observed similar distribution of cell cycle stages upon continuous treatment with xenoestrogens. I guess my question is if the number of cells with more than two centrosomes change with passaging of cells in presence or absence of estrogens, and how? The authors should discuss possible mechanisms and implications or perform new experiments similar to those in Fig. 1 but with few time points at which they could follow the proportion of cells with CA.

We thank the referee for important comments and suggestions. We now discuss all of these points in the revised manuscript and implemented important further experiments.

The referee is correct that a p53-mediated cell cycle arrest and proliferation deficits would have to be assumed at the first glance. However, some points should be considered: We observed very low levels of CA in our colon (cancer) cell systems (i.e., < 10%). The study by Holland et al demonstrates a p53-dependent proliferation deficit in cells with a rapid and highly penetrant CA (>85-95%). Importantly, the amount of cells with extra centrosomes declined to values <10% due to these proliferation deficits in the cells harboring extra centrosomes. We predict that a larger proportion of cells with CA will be detrimental to long-term survival. This fits with our data illustrating a constant low level of CA below 10% over 30 generations of estrogen-treatment (Figs S1C-D, 5E-F and S5D-G) that did not apparently effect cell cycle distribution (Fig 4G) or cell proliferation (Fig 3G-H). The low hormone concentrations used in this study are important in this context because it has been previously reported that higher doses in the micromolar range suppress microtubule polymerization and dynamics or even disrupt the microtubule network and arrest cells in mitosis, possibly by mechanisms similar to spindle poisons (Metzler et al, 1995; Pfeiffer et al, 1997; Brueggemeier et al 2001; Nakagimi et al, 2001; Kamath et al, 2006; Jurasek et al, 2018). By contrast, nanomolar hormone-concentrations did not arrest cells in mitosis (Fig 4G). This is important because mitotic arrest can be lethal in several ways (summarized in [Bekier 2009]), counteracting continuous chromosome mis-segregation that manifest in w-CIN and aneuploidy. Furthermore, previous studies found that not all an-

euploid cells (RPE1) resulting from erroneous mitoses activate p53 (Santaguida et al., 2017; Soto et al., 2017). Thus, the p53-mediated G1 arrest is a potential but not obligatory outcome of aneuploidy and may reflect more complex aneuploidy or structural aneuploidy than numerical defects. We edited the previous discussion and now included most of these arguments (see p.27).

The referee's suggestion to follow centrosome number evolution is excellent, as this would point out whether the level of CA would increase over time or remain constant, possibly by losing extra centrosomes (Ganem et al., 2009). We therefore have performed additional experiments including further time points of estrogen-treatment (2, 4, and 6 days). The results from these data are presented in new Figs S1C and D, illustrating a constant low level of CA over time that was comparable to those that we measured after 48 h. Importantly, absolute levels of CA remained constant, even if concentrations in the micromolar range were used (current Figs S1A-B and S2 H-I, cross-comment of referee #2). This is in line with non-monotonic concentration-effect relationships in which increasing hormone-concentrations do not result in increased effects across the entire concentration range (Beausoleil et al., 2013, Vandenberg et al., 2012). Interestingly, continuous estrogen-treated w-CIN clones also show total levels of CA below 10% after 30 days (Fig 5E and F and S5D-G) (cross comment of referee #3), which not only proves the direct relationship between CA and w-CIN (Ganem et al., 2009), which has not yet been demonstrated in a colon (cancer) system. Our data also imply that the generation of just one extra centrosome per cell (i.e., 3 centriole positive centrosomes) and overall low frequencies of CA are sufficient for w-CIN development. By contrast, large numbers of supernumerary centrosomes per cell with high frequencies are likely to adversely affect cell viability because extra centrosomes cluster inefficiently during mitosis and increase the frequency of lethal multipolar divisions [3]. In line with Levine et al., creating just one or two extra centrosomes per cell via modest increased Plk4 levels, we agree with the authors' assumption that large numbers of centrosomes per cell and overall high frequencies of CA will be detrimental to long-term cell survival (Levine et al, 2017). This is consistent with our results demonstrating unaffected cell proliferation in the presence of (xeno)estrogens within 7 days of treatment (Fig 3G and H), and is further supported by Holland et al. (see above).

We have now discussed these points in detail in the revised version of the manuscript (see p. 28).

2) In relation to the previous comment, the authors observed a low percentage of cells with centrosome amplification (max. 10%) even after prolonged treatment with inhibitors, and the authors do not describe the reason for this.

The percentage of cells with CA is low (i.e., <10%, Fig 1) but comparable to data from previous studies detecting ~10-15% of cells with amplified centrosomes after exposure with low estrogen-concentrations (Ho et al, 2017; Kim et al, 2009; Tarapore et al, 2014). Even overexpression of PLK4, a key regu-

lator of centrosome duplication (Habedanck et al, 2005), causes only a slightly higher proportion of cells with CA (Fig S1E, Ganem et al, 2009), suggesting a threshold above which a growth disadvantage of cells with CA should be suspected (see above).

Of note, supernumerary centrosomes lead to the formation of multipolar spindles, which cause multipolar cell divisions if not corrected otherwise. However, multipolar mitoses are likely to be detrimental to cells due to gross chromosome missegregation and cell death (Ganem et al, 2009). (Xeno)estrogens trigger the formation of supernumerary centrosomes and as expected, multipolar mitotic spindles (Fig 1 and 4C). However, we did not apparently observe multipolar mitoses and cell death in live-imaged cells that were exposed to (xeno)estrogens (Fig 4A and S4C). Importantly, (cancer) cells can “cope” with extra centrosomes by several mechanisms to limit the detrimental consequences of CA (Godinho et al, 2009). We hypothesize, that (xeno)estrogen-treated cells likely avoid lethal divisions by coalescence of extra centrosomes to form a pseudobipolar spindle. Consistent with this, (xeno)estrogen-treated cells display an increased frequency of lagging chromosomes during anaphase (Fig 4E-F and S4G-H), which typically arise from multipolar spindle intermediates and merotelic kinetochore attachments (Ganem et al. 2009). Not least, as seen for a moderate increase in Plk4 (Levine et al, 2017), (xeno)estrogens induce the creation of just one extra centrosome per cell, which is permissive for centrosome coalescence and cell survival. By contrast, large numbers of supernumerary centrosomes are likely to be detrimental to cell viability due to inefficient clustering prior to division. We have adapted the discussion in order to create more clarity (p. 27-28).

This is followed by a similarly low percentage of cells with lagging chromosomes and increased mitotic durations, implying that cells that did not show an increased number of lagging chromosomes and increased mitotic durations are those that have a normal number of centrosomes. An alternative explanation is that for some unknown reason certain cells do not respond to this treatment, but that is probably not the case, as only a slightly higher percentage of CA cells is observed after overexpression of PLK4 (S1C). If time permits, the authors could synchronize the cells before treatment to enable synchronous passage of the population through the S phase, possibly on only one representative cell line. In this way, the authors could observe whether xenoestrogens act on all cells in S phase equally or whether there are other factors that are more essential to maintain the number of centrosomes.

We agree with the referee that it is rather unlikely that certain cells did not respond to the estrogen-treatment, especially, as low levels of CA were similarly observed after overexpression of PLK4 (current Fig S1E). In line with this, the low levels of lagging chromosomes observed upon estrogen-treatment is similar to previous studies illustrating 6-7% of anaphase cells possessing mitotic laggards upon the treatment (Stolz et al., 2010, Ertych et al., 2014, Stolz et al., 2015). Of note, only a slightly higher percentage of lagging chromosomes is observed in chromosomally unstable cell lines (Bakhoum et al., 2014).

As suggested by the referee, we performed further experiments to clarify possible mechanisms of CA in response to estrogens. Synchronization of cells at G1/S following estrogen-treatment allowed synchronous passage of the whole cell population through the S-phase (novel Fig 2F-I, and S2D). Instead of quantifying centrosome numbers in the next cell cycle, we decided to measure Sas-6 fluorescence intensities directly at the time of treatment, i.e., during S-phase, to avoid interfering influences from other cell cycle phases. In line with Kim et al., we observed an increase in Sas-6 fluorescence intensities at S-phase centrosomes (novel Fig 2G-I), suggesting that estrogenic substances disturb the centrosome cycle by triggering centriole overduplication (Kim et al., 2009). In contrast, synchronization of cells following estrogen-treatment from G2 to M phase did not result in unequal distributed centrioles among cell poles (novel Fig S2E-G) collectively supporting centriole overduplication but not premature centriole disengagement at early mitosis as the underlying mechanism of CA.

Also, authors should quantify the proportion of cells with transient multipolar H2B phenotype and unaligned chromosomes during pseudo-metaphase (see minor comment 4) in live-cell imaging experiments after treatment with xenoestrogens, now only represented as images on Fig. 3A. This could be a good indication of how effective treatment is and whether cells that showed strong delays in mitosis are those that have transient multipolar spindles that are connected with amplified centrosomes (Ganem et al., 2009, Nature). *Quantifications as suggested have been performed and the results, demonstrating an increase in transient multipolar and unaligned chromosomes upon estrogen-treatment, are added in current Fig S4D-E. Given that chromosome alignment defects occur during pseudo-metaphase, this phenotype can be most accurately detected in cells synchronized at metaphase, i.e., to distinguish prometaphase cells with not yet fully aligned chromosomes, which is normal at this state, from pathophysiologic pseudo-metaphases. Panel D of current Fig 4 illustrates the results of this experiment, verifying data from live-imaged cells. Note that the absolute number of pseudo-metaphases are lower under this experimental setup, likely due to a smaller amount of false positives. To investigate whether extra centrosomes drive these phenotypes, we re-quantified the multipolarity phenotype with centrosome and microtubule markers as well, as suggested by referee #3. The results of these experiments are included in the new panel C in Fig 4, supporting our results from live-imaged cells and additionally demonstrating the link between strong mitotic delays of single cells, supernumerary centrosomes, and transient multipolar spindles, as shown by Ganem et al., (Ganem et al., 2009).*

The authors should also indicate that the increased number of unaligned chromosomes could be a possible cause of the increase in the number of lagging chromosomes (see Vukušić&Tolić, 2022, Cells).

We agree with the referee and consider this important paper, as it points out the link between extra centrosomes, chromosome mis-alignment and lagging chromosomes (see page 15, line23).

Minor comments:

1) In the Introduction, page 1, line 26 the authors state that lagging chromosomes randomly segregate to new daughter cells. This is only partially true as there are increasing number of papers that reported non-random inheritance of lagging chromosomes that is influenced by age of centrosome in daughter cells (Gasic et al., 2015, eLife), activity of Aurora B within spindle midzone (Sen et al., 2021, Developmental Cell; Orr et al., 2021, Current Biology), or number of microtubules attached to each sister and midzone-derived forces during anaphase B in general (Cimini et al., 2004, Current Biology). Authors should acknowledge the mentioned papers or change this sentence.

We thank the referee for this clarification and rephrased the mentioned sentence as follows: "It is widely accepted that lagging chromosomes represent an important mechanism for whole chromosomal instability (w-CIN) and aneuploidy..." (see p. 3, line 28-30)

2) In a sentence on page 3, line 24 ('Cells with...'), the authors should cite the fundamental work of the David Pellman group linking centrosome amplification, transient multipolar spindles, and lagging chromosomes in human cells (Ganem et al., 2009, Nature).

Of note, we mentioned this fundamental work of the David Pellman group within the manuscript at several sites (on page 14, line 45; page 15, line 23; page 19, lines 9 and 14; page 27, lines 8 and 26; page 28 line 1). However, we fully agree with the referee and have now included the Ganem et al. reference in the introduction as well when introducing the link between CA, transient multipolarity, and lagging chromosomes in human cells (page 3, line 28).

3) Regarding the observation that CA increased mitotic duration, the authors should cite the fundamental work from the Conley Rieder lab, since it showed that an increase in the number of centrosomes increases the duration of mitosis in normal human cells irrespective of other changes (Yang et al., 2008, Nature Cell Biology).

We have now included the Yang et al. reference on page 14, line 37 and thank the referee for his remark.

4) On page 11, line 43 the authors use the term 'prometaphase-like state' to refer to metaphase cells with unaligned chromosomes. I would advise the authors to use the term 'pseudo-metaphase', a widely accepted term for such mitotic cells, as defined by Weaver et al., 2003, J Cell Biol.

We have followed the referee's advice and rephrased the term "prometaphase-like state" to "pseudo-metaphase" whenever we referred to metaphase cells with unaligned chromosomes (page 14, lines 42-43; page 15, line 8). We also adapt the corresponding figures 4D and S4E.

5) On page 11, line 38, the authors state that cells treated with xenoestrogens 'progressed through mitosis and segregated their chromosomes, albeit with reduced fidelity'. The term reduced fidelity should be much more precise. I would suggest that the authors use the term 'unaligned chromosomes,' as they did in the figure caption, which are only transiently present in treated cells during pseudo-metaphase in this study, as such chromosomes could drastically reduce the fidelity of mitosis.

The referee is correct. We have now reworded the sentence by replacing the term 'reduced fidelity' with a more precise phrase: "...progressed through mitosis and segregated their chromosomes, although alignment defects, reminiscent of a "pseudo-metaphase" (Weaver et al., 2003) were detected." (see page 14, lines 42-43).

6) I would encourage the authors to include additional examples of live-imaged mitotic cells in Fig. S3, similar to those presented on Fig. 3A. I would also encourage authors to enlarge the live-cell H2B figures, as currently they are in a too small format.

We have implemented the referee's comments and included additional examples of live-imaged mitotic cells in the revised manuscript (current Fig S4C) and enlarged existing images as suggested (Fig 4A). In addition, we included further Videos representative for the respective mitotic defect (see Supplementary Videos S4-8).

7) I did not understand why authors did not quantify the proportion of cells with lag-gards/chromosome bridges from live-cell movies of treated, and untreated cells. If there is some specific reason for this, please include it in the Results section, otherwise authors should quantify the proportion of such chromosomes in a population of imaged cells.

Lagging chromosomes are kinetochore-positive chromosomes that lag in between the segregating chromosomes during anaphase due to kinetochores, which are concurrent attached to microtubules emanating from both spindle poles (merotelic orientation) (Cimini et al., 2001). Live-cell imaging performed in this study was performed by using H2B-GFP to visualize chromosomes, but a kinetochore marker was lacking. Apparent chromosome fragments observed in live-imaged cells could therefore not be assigned to lagging chromosomes with certainty. Therefore, we quantified the amount of anaphase lagging chromosomes in fixed cells by using CREST as a kinetochore marker (current Fig 4E-F and S4G-H). Nevertheless, we followed the referee's instructions and quantified the amount of cells with chromosome bridges from live-cell movies (Fig S4F) because this phenotype can be clearly identified.

8) On page 12, line 1 the authors should cite the work from Daniel Gerlich lab that showed how increased mitotic duration observed after treatment of cells with low concentrations of nocodazole scales with the number of unaligned chromosomes (Dick&Gerlich, 2013, Nature Cell Biology).

We are thankful for this remark and now included the Dick&Gerlich reference on page 15, line 3 and thank the referee for his remark.

9) In the caption of Figure 3C, include the exact concentration of nocodazole used, as it is critical to the observed degree of mitotic delay after this treatment (see previous comment).

The exact concentration of nocodazole used, i.e., 5 nM, was already included in the caption of previous Fig 3A from which previous figures 3B and C were extracted from. To make this more clear, we rephrased the caption of current Fig 4 as follows: “(B) The time from nuclear envelope breakdown (NEB) to anaphase onset was determined from (xeno)estrogen [10 nM each] and Noc-treated [5 nM] cells...” Since we transferred previous Fig 3C to the Supplemental Fig S4B, we included a new caption for this panel and referred to Fig 4B (see p. 56).

10) It is unclear to me from the caption of Fig. 3A and the M&M section "Immunofluorescence microscopy" whether the cells were live imaged in the presence of xenoestrogens. Please rephrase the sentences to make this clearer or introduce a scheme of experimental design in Fig. 3A or Fig. S3.

We apologize for this confusion. We now included a scheme in Fig S4A illustrating the experimental setup for current Fig 4A, B and S4B-F. We additionally rephrased the caption for current Fig 4A: "...following live cell imaging for 8 h under continued treatment." (page 18).

11) Related to the major comments presented above, the authors should discuss why the number of aneuploid cells observed in long-term treatments that span 30 generations (Fig. 4) is much larger than the proportion of cells with CA (Fig. 1), and cells with lagging chromosomes (Fig. 3) after short treatments. Related to that point, what would authors expect if cells were treated with estrogens for a short time period and then left for 30 generation? The authors should discuss implications and possible mechanisms of coping with centrosome amplification in this context.

We now discuss in detail how cells “cope” with extra centrosomes (associated with the formation of lagging chromosomes) to avoid the detrimental consequences of multipolar mitoses (see p. 27, line 23 – page 28, line 8), and the observed karyotype variability of 30-40% (page 28 line 18 – page 29, line 6). We suggest that cells have evolved several mechanisms to cope with extra centrosomes to keep the level of CA (and thus also of lagging chromosomes) as low as possible to ensure cell survival. By contrast, cells with complex karyotypes will reach an “optimal” level with maximized selective advantages of distinct chromosomal aneuploidies (Madhwesh et al. 2018); and optimal rates of chromosome missegregation probabilities might exist that maximizes karyotypic heterogeneity (Elizalde et al. 2018).

Regarding the latter question, we have included a separate paragraph discussing our expectation regarding the evolution of centrosomes after recovery from (xeno)estrogen treatment (see page 28, line 8-17).

12) Cite appropriate papers for sentence on page 19, line 3 ('Our results hypothesize...').

We reconsidered the text passage and found it misleading. According to our current knowledge, our study is the first showing a connection between estrogen-exposure, extra centrosomes and w-CIN in the colon (cancer) system.

Therefore, we edited the text passage as follows: “Our results hypothesize a potential cause of amplified centrosomes and w-CIN in CRC cells that may include exposure to certain environmental estrogenic substances” (see page 25, lines 5-6).

13) The authors should further discuss the implication of their observation that CRC-derived and normal colon epithelial cells behave similarly after xenoestrogen treatment in this study, except the fact that percentages of cells with CA and lagging chromosomes are lower in normal colon cells. Interestingly, the amount of aneuploidy observed in the population is similar after long-term treatment with xenoestrogen in HCT116 and CCD 841 CoN cells.

We used non-transformed cells to uncouple (xeno)estrogen-mediated effects from cell degeneracy. The similar behavior of transformed and non-transformed cells might thus mirror cell transformation-independent effects as previously mentioned on p. 5, line 30 and currently, on p.28, line 21. Therefore, we had already made the following implication in the previous manuscript (now, slightly edited): “Thus, our data suggest that exposure to GPER1-activating estrogenic substances, such as E2, BPA, and DES may promote genomic instability in intestinal cells that might could persist during CRC progression.” (see page 19, lines 2-4).

Regarding the letter comment, we suggest, that the amount of aneuploidy observed in transformed and non-transformed cell lines may not yet be exhausted. Thus, it could be that the levels between the two cell lines could differ with further passaging. We now refer to an in silico study by Elizalde et al demonstrating that karyotype diversity is significantly more dependent on the chromosome missegregation rate than on the number of cell divisions. Thus, karyotype diversity can be reached rapidly at high missegregation rates. By contrast, at low missegregation rates karyotype diversity is expected to be constrained after more cell division events. With respect to non-transformed cells used in our study exhibiting lower rates of lagging chromosomes (CCD 841 CoN, 2.5-3%) compared to transformed HCT116 cells (~5%) (Fig. 4E and F), this could mean that maximal karyotype heterogeneity might not be exhausted in both cell lines after 30 days of (xeno)estrogen treatment, but even more in non-transformed cells (see page 28, line 32 – page 29, line 6).

14) Label treatments in the left part of Figure S3D.

We thank the referee for the attentive look. We have added the missing label of treatments in the mentioned figure that now switched to Fig 4G.

15) The 'Vinca alkaloid' is mentioned on page 12, line 1, but there are no data for it.

Data for the Vinca alkaloid nocodazole were already included in previous Fig 3C (green column as usual) and are now switched to Fig S4B.

16) On page 12, line 7, thymidine is misspelled as thymine.

We have corrected this spelling mistake accordingly and thank the referee for pointing it out.

17) Figure S1D could be placed in a main figure, as it implicated PLK4 as a bottom target of estrogen-GPER1 signaling, which is essential for centrosome amplification.

Previous Fig S1D is now shifted to the main figure 2, panel E. Together with new results presented in current Fig 2F-I, illustrating centriole overduplication as a possible underlying mechanism of (xeno)estrogen-induced CA, Plk4, i.e., the master-regulatory kinase of centriole duplication, becomes a bottom target of estrogen-GPER1 signaling in context of CA.

18) Graphical cover should not be depicted as a circle as it is not a feedback loop since chromosomal instability does not lead to GPER1 activation.

We apologize for this misleading. We have edited the graphical abstract accordingly.

19) The authors should calculate error bars and statistical significance of results presented on Figures 4C, 5C, 5D, S4B, and S5D.

Data presented in the above mentioned figures (now shifted to Figures 5C, 6C, 6D, S5C, and S6F) illustrate the results from karyotype analyses of cell populations treated for 30 days. This is different to Fig 5A where statistics of several independent cell clones were calculated. Analyses of cell populations make the calculation of p-values quite difficult, as biological replicates are missing. Analyses of whole cell populations was preferred to individual clones when this is dictated by (i) the cell line (e.g., non-transformed cells that do not grow in the absence of cell-cell contact, obligatory needed to generate single cell clones), or (ii) a complex experimental setup (e.g., high number of different treatments). Even though we were not able to provide the p-values, we tried to accommodate the expert by now providing the standard deviations of the values.

Reviewer #2 (Comments to the Authors (Required)):

Buehler, Stolz and colleagues here explore the impact of (xeno-)estrogens on centrosomes and chromosome stability in colon cancer cells and colon epithelial control cells. They demonstrate the induction of centrosome abnormalities by GPER1-activating estrogen treatment that are accompanied by mitotic problems and chromosome instability. These data are of potential relevance in how estrogens may contribute to cancer development in the colon. The findings are novel and should interest researchers in a several fields.

In general, the experiments in the paper are thoroughly done and well described; the key conclusions of the study are supported by the data presented.

We are pleased that the data presented in the manuscript, as well as their description and conclusions, generally convinced the referee.

I suggest some additional control experiments and clarifications that should be included to strengthen the findings presented here:

1. A frequency distribution of the centriole numbers should be provided, based on counts with CEP135 or CP110. Without any detectable impact on the cell cycle distribution of the treated populations, it should be tested whether the treatment outcomes are due to loss of centriole cohesion or CA. Related to this point, do increasing doses of estrogens cause increasing centriole numbers?

This is an excellent suggestion. We have now included a frequency distribution of centrosome and centriole numbers, based on the quantification of γ -tubulin, Cep135 and CP110, respectively. The results, illustrating the occurrence of mainly three (and not four or more of four) centrosomes, are included in current Fig 2A-D and S2A-C. We followed the referee's comment and examined, whether centriole cohesion, i.e., premature centriole disengagement during early mitosis, or CA originating from centriole overduplication represents an underlying molecular mechanism. The new results were added to Fig 2F-I and S2D-G demonstrating increased Sas-6 fluorescence intensities at S-phase centrosomes, which supports overduplication. However, given that overduplication involves either de novo assembly of centrioles or parental centrioles templating the assembly of more than 1 centriole each, it is difficult to point out the exact molecular mechanism at this point. Further exciting analyses will be required here in the future.

Concerning increasing concentrations of estrogens, we did not observe an obvious increase in centriole overduplication after (xeno)estrogen-exposure compared to the control (new panels H-I in current Fig S2). The different outcomes are in line with non-monotonic concentration responses that follow unpredictable or atypical patterns, in which increasing doses do not result in increased effects across the entire concentration range (Beausoleil et al., 2013, Vandenberg et al., 2012).

2. A rescue control for the siRNA experiments in Figures 2 and 5 should be provided.

We followed the referee's suggestion and performed GPER1 siRNA rescue experiments to demonstrate the dependency of CA (new Fig S3G-H) or lagging chromosomes (new Fig S6A-B) on GPER1, and the specificity of the GPER1 knockdown (new Fig S3F). Nevertheless, we feel that the GPER1-dependence of the observed estrogen-responses is well demonstrated by the use of GPER1-specific agonists (current Fig 3E and F), antagonists (current Figs 3C and D, S3K, S6C, D, F, and G), and well-known GPER1-activators (current Fig S3J and L).

A rescue control for w-CIN and aneuploidy data was not performed for technical reasons because the siRNA resistant plasmid harboring the silent mutations for GPER1 would have had to be co-transfected every two to three days to guarantee adequate expression levels, which would have been very stressful for the cells. Otherwise, we would have had to generate completely new GPER1 siRNA resistant cell clones and ensure continuous GPER1 knock-down via continuous siRNA transfections, with the same result. For these reasons, we decided against these experiments.

3. The data in Fig. S3D are particularly important and should be included in the main manuscript. These data exclude the possibility that an estrogen-regulated cell cycle arrest causes the centrosome phenotype. The information in this Figure should be improved: the 4 panels in the FACS plot should be labelled for their respective treatments and the description of the cell cycle phases should reflect those in the Figure itself -here are 4 categories of cell in the graph, but only 3 mentioned in the legend; and it seems that sub-G1 cells are not included in the graph (?). It should be stated how many times this experiment was repeated and relevant statistics included in the Figure.

The data in previous Fig S3D were now transferred to the main Figure 4, panel G in the revised manuscript. We thank the referee for his comments and attentive look. We have added the missing labels on the left panel and improved the information in the Figure according to the referee's suggestions. The right part of the figure is now larger so that the individual bars reflecting the cell cycle phases can be seen more clearly. The labels have been adjusted ("G2/M" instead of "G2"). Previously, a representative example was shown in Fig S3D. We have now merged all three independent experiments and included relevant statistics.

Cross comments on referee reports:

I agree with the other 2 reviewers' point on the need for a more extended timecourse of centrosome numbers over treatment time. This (and the related interpretation of the results, of course) would provide important new information. I suggest that the authors count centrioles during this timecourse; this point was also raised by referee 3.

We followed the referee's cross comment on referee #1 to follow CA over time. We included an extended time course of centrosome numbers over 2, 4, and 6 days, covering the time of cell proliferation analysis (Fig S1C and D). In addition, we quantified the amount of cells with supernumerary centrosomes in cells, which were long-term treated with estrogens for 30 days, and used for karyotype analyses (Fig 5E and F and S5D-G), to show centrosome number evolution over time. Results from these data illustrate a constant maximum level of CA <10% during the time course, comparable to our previous results of a 48 h-treatment (Fig. 1) and similar to transient overexpression of PLK4 (Fig S1E), Ganem et al. 2009). This low amount of cells with supernumerary centrosomes did not increase further by using higher concentrations in the micromolar range (current Figs. S1A-B and S2H-I). The results of these data suggest that:

- i) Nanomolar hormone concentrations induce low levels of CA, that do not apparently interfere with cell cycle distribution or cell proliferation, unlike cells with high levels of CA (>85-95%) (Holland et al., 2012);*
- ii) Centrosome numbers were stable within 30 generations supporting a direct link between CA and w-CIN (Ganem et al., 2009);*
- iii) A maximum level of CA that is <10% seems to be compatible with cell proliferation, which is in line with the study of Holland et al., illustrating a decline*

in cells with strong CA to <10% by 10 days after PLK4 overexpression (Holland et al., 2012);

iv) Hormones follow non-monotonic concentration–effect relationships in which increasing doses do not result in increased effects across the entire concentration range (Beausoleil et al., 2013, Vandenberg et al., 2012).

Concerning centriole numbers, we have now included a frequency distribution of centrosome and centriole numbers, based on the quantification of γ -tubulin, Cep135 and CP110, respectively. The results, illustrating the occurrence of mainly three (and not four or more of four) centrosomes, are included in current Fig 2A-D and S2A-C. We examined whether centriole cohesion, i.e., premature centriole disengagement during early mitosis, or CA originating from centriole overduplication represents an underlying molecular mechanism. The new results were added to Fig 2F-I and S2D-G demonstrating increases Sas-6 fluorescence intensities at S-phase centrosomes, which supports overduplication. However, given that overduplication involves either de novo assembly of centrioles or parental centrioles, which template the assembly of more than 1 centriole each, it is difficult to point out the exact molecular mechanism at this point. Further exciting analyses will be required here in the future.

It may be challenging to synchronise the cells (as suggested by referee 1) without impacting on centriole numbers. The authors should consider carefully how to approach this suggestion.

The referee is correct with his concerns. However, by using control-treatments in parallel, we feel to allow adequate assessment of the results.

Reviewer #3 (Comments to the Authors (Required)):

In this manuscript the authors demonstrate that in normal and cancerous colonic epithelial cells estrogen-dependent stimulation of the G protein coupled estrogen receptor GPER1 is responsible for supernumerary centrosomes, aneuploidy and an increased variability in mitotic duration. Intriguingly, estrogens via GPER1 seem to trigger chromosomal instability at levels comparable to that caused by low doses of nocodazole. Overall the data on the role GPER1 in karyotype stability is convincing, however, additional experimental data is needed to show if this is indeed due to centrosome amplification.

We are pleased that our data generally convince the referee.

1. The centrosome amplification phenotype shown in Fig 1 and Fig S1 is interesting, although it is fairly infrequent in the population (remains under 10% despite 48 hr treatment). My concern is that all the images (throughout the manuscript) depict precisely three PCM-containing centrosomes (which means that these centrioles would have been produced in the preceding cell cycle or before). Normally, overduplication involves either de novo assembly of centrioles, or parental centrioles templating assembly of more than 1 centriole each. It is difficult to imagine that such pathways yield 3 centrosomes.

In my opinion the phenotype the authors see could be due to unequal segregation of centrosomes between daughter cells with one cell inheriting three and the other a single centriole.

Such a phenotype arises if a parental centriole and its procentriole split apart (i.e. disengage prematurely) during mitosis and the four centrioles segregate unevenly. These split centrosomes may even promote formation of additional spindle poles, so the transient multipolarity the authors allude to would fit this model. Moreover, a mitotic delay can trigger centrosome splitting and abnormal centriole/centrosome segregation, and thus the centrosome phenotype may actually be secondary to a mitotic defect. To distinguish between these possibilities the authors should score centrosome numbers more precisely (i.e. whether cells contain 0, 1, 2, 3, 4 or more centrosomes). If it is confirmed that many cells have more than 3 centrosomes, then trying a procentriole marker like Sas6 will help establish if parental centrioles indeed assemble more than one centriole each and if de novo assembly of procentrioles occur.

We are most thankful for the valuable comments and suggestions made by the referee, which fits to a cross comment by referee #2. We have now included a frequency distribution of centrosome and centriole numbers, based on the quantification of γ -tubulin, Cep135 and CP110, respectively. The results verify the occurrence of mainly three (and not four or more than four) PCM-containing centrosomes as previously shown in the images. The data of these results are included in current Fig 2A-D and S2A-C. We further followed the referee's comment and examined, whether centriole cohesion, i.e., premature centriole disengagement during early mitosis, or CA originating from centriole overduplication would represent an underlying molecular mechanism. The new results were added to Figs 2F-I and S2D-G demonstrating increased Sas-6 fluorescence intensities at S-phase centrosomes, supporting an overduplication mode of action. However, given that overduplication involves either de novo assembly of centrioles or parental centrioles, which template the assembly of more than 1 centriole each, it is difficult to point out the exact molecular mechanism at this point. Further exciting analyses will be required here in the future.

2. The multipolarity phenotype should be shown with centrosome and microtubule markers as well to establish if extra centrosomes are driving this phenotype.

We followed the referee's suggestion and now demonstrate the multipolarity phenotype by staining the cells with antibodies against α - and γ -tubulin, to visualize microtubules and centrosomes, respectively. We included the data from these results in the new Fig 4C.

3. It would be crucial to show centrosome number evolution in the clones that were used for karyotype analysis following 30 generations.

As suggested by the referee, we now demonstrate the proportion of cells with supernumerary centrosomes in the (xeno)estrogen-treated clones. The results from these data are presented in new Fig 5E-F, and S5D-G, illustrating a constant low level of CA over time that was comparable to those that were measured after 48 h. These data not only support a direct link between estrogen-induced CA and w-CIN (Ganem et al., 2009), but also suggest a maximum level of CA <10% that is inducible by (xeno)estrogens and not apparent-

ly affecting cell cycle distribution or cell proliferation, which is in line with studies of Holland et al., 2012.

The low level of CA (<10%) observed in the different cell lines are in range with studies of Tarapore et al., 2014, Ho et al., 2017 and Kim et al. 2019 also illustrating ~10-15% of cells with amplified centrosomes after exposure with low doses of estrogens. As mentioned above, we suggest that this might be the maximum level of CA, as neither long-term treatment of cells nor higher concentrations of estrogens in the micromolar range result in higher percentages of CA (current Figs S1A-B, S2H-I, 5E-F, and S5D-G).

4. It is perplexing that all the treatments including nocodazole give rise to a nearly identical level of aneuploidy (30-40% of cells) both in HCT116 and normal colon cells. Could it be that this is the limit at which the cells can still proliferate?

The reviewer asks a very interesting question, which is similar to the cross comment of referee #1. It was shown that cells with complex karyotypes will reach an “optimal” level with maximized selective advantages of distinct chromosomal aneuploidies (Madhwesh et al. 2018). By contrast, higher levels could indeed have detrimental consequences to cellular fitness. On the other hand, the amount of aneuploidy observed in transformed and non-transformed cell lines may not yet be exhausted, meaning that the levels between the two cell lines could differ with further passaging. We now refer to an in silico study by Elizalde et al demonstrating that karyotype diversity is significantly more dependent on the chromosome missegregation rate than on the number of cell divisions. Thus, karyotype diversity can be reached rapidly at high missegregation rates. By contrast, at low missegregation rates karyotype diversity is expected to be constrained after more cell division events. With respect to non-transformed cells used in our study exhibiting lower rates of lagging chromosomes (CCD 841 CoN, 2.5-3%) compared to transformed HCT116 cells (~5%) (Fig. 4E and F), this could mean that maximal karyotype heterogeneity might not be exhausted in both cell lines after 30 days of (xeno)estrogen treatment, but even more in non-transformed cells (see page 28, line 32 – page 29, line 6).

October 18, 2022

RE: Life Science Alliance Manuscript #LSA-2022-01499-TR

Dr. Ailine Stolz
Federal Institute for Risk Assessment
Experimental Toxicology and ZEBET
Max-Dohrn-Str. 8-10
Berlin 10589
Germany

Dear Dr. Stolz,

Thank you for submitting your revised manuscript entitled "GPER1 links estrogens to centrosome amplification and chromosomal instability in human colon cells". We would be happy to publish your paper in Life Science Alliance pending final revisions necessary to meet our formatting guidelines.

- please address Reviewer 2 and 3's remaining comments
- please use the [10 author names, et al.] format in your references (i.e. limit the author names to the first 10)
- please add a separate figure legend section to your main manuscript text
- please incorporate your supplementary methods section into the main methods section; we do not have a word limit on this section
- please incorporate the Supplemental References into the main Reference list
- the short section entitled "Supplementary Information", listed after the Data Availability section, can be removed

A. FINAL FILES:

B. MANUSCRIPT ORGANIZATION AND FORMATTING:

Sincerely,

Reviewer #1 (Comments to the Authors (Required)):

The revised manuscript "GPER1 links estrogens to centrosome amplification and chromosomal instability in human colon cells" by Buhler et al. is, in my opinion, a significant improvement over the first version of the manuscript originally submitted to the journal. The authors have done a great job in responding to the reviews' comments, resulting in the expansion of conclusions by introducing a lot of new experimental data, reanalyzing existing data, reorganizing the figure panels, introducing more relevant movies, and especially significantly improving the discussion of possible mechanisms of centrosome amplification after treatment with xenoestrogens. The authors also nicely present possible mechanisms by which cells cope with excess centrosomes. In the revised manuscript, the authors introduced new experiments where they followed the proportion of cells with CA over time, new quantification of centrosomes with specific markers Cep135, Cep110, and Sas6, new quantification of mitotic abnormalities with kinetochore markers, and new analysis of previously acquired data. Together, the new data greatly strengthens the main conclusions of the manuscript. In my opinion, the manuscript presented in this way would be a valuable contribution to the field and therefore I strongly advocate the publication of the manuscript.

Referee Cross-Comments

I have no objections to the comments of the other two reviewers.

Reviewer #2 (Comments to the Authors (Required)):

Buehler, Stolz and colleagues have significantly revised their study of xeno-estrogen-induced centrosome amplification (CA). They have addressed experimentally the points raised in my initial review and have provided new data in an improved manuscript that one expects to interest several different readerships. The main findings here are convincing and novel and I am supportive of the publication of this study. However, while I do not suggest any additional experiments, I have some points that I consider should be addressed beforehand.

1. p6 L14: The detailed tables of centrosome/ centriole numbers in Fig. S2A-C do not support the statement, 'The frequency distribution of centriole numbers based on counts with Cep135 and CP110 (Fig S2B and C) was almost identical to that with gamma-tubulin (Fig 2A-D and S2A), thereby confirming the formation of exactly three centriole-positive centrosomes upon (xeno)estrogen treatment.'

a. That there is still a population of cells with >4 centrioles after some treatments in the CRC lines argues for caution in asserting formation of 'exactly' three centriole-positive centrosomes. As the mechanism of CA caused by the xeno-estrogens remains to be clarified, it is not ideal to assert a very specific outcome to the treatments in terms of centrosome numbers; this is potentially misleading and should be toned down. One possible consideration is that the low number of cells with 4 centrioles and the flow cytometry data would suggest that most CA is happening prior to the completion of normal centriole duplication, so that there is a

reduced likelihood of >4 centrioles being formed.

b. Given the relatively low overall levels of CA observed, I conclude from the data presented that there are distinct differences seen using gamma-tubulin and CEP135 or CP110. Whether these differences are biologically significant is outside the scope of this paper, but the authors should revise their interpretation of these data.

2. The experiment to explore centriole 'disengagement' presented in Fig. S2E examines loss of centrosome separation at M phase onset, rather than disengagement, which was the term used in the work from the Stearns lab for the licensing of centriole duplication. This should be rephrased to avoid confusion.

3. The data in the Table in Fig S2A are presented in a bar chart form in Fig. 2A-D (one presumes these are the same data). This redundancy is not ideal. I suggest that replacing the bar charts with the tabular data would reduce the amount of supplementary data and make the main MS. more complete.

4. (Minor point) p4 'Evovement' should be replaced by 'evolution'.

Reviewer #3 (Comments to the Authors (Required)):

The authors have performed several new experiments that greatly improved the manuscript. At this stage the paper would benefit from a thorough edit as some of the sentences are long and cumbersome, making it harder to follow the logic of the paper.

1. Please rephrase this sentence, which is a good example of the above:

However, because Plk4 plays a central role in regulating centrosome numbers, we cannot exclude the possibility that its superordinate role overlays other underlying mechanisms of (xeno)estrogen-triggered CA.

2. The centriole disengagement experiment is described in a rather confusing manner and the illustration in Fig S2E is not particularly helpful. Premature disengagement leads to procentrioles (lacking PCM) being freed from their parent. Such procentrioles would lack their own PCM and would not associate with their parent's PCM either (as depicted in illustration). If Sas6 coincides with PCM markers in all cells, then premature disengagement is indeed unlikely, but a picture demonstrating these results could accompany the graphs in F and G.

3. The authors state: At the end of mitosis, the parental centriole and its overduplicated procentrioles split apart (i.e. disengage), resulting in daughter cells with three centrioles. It may be three or five centrioles. There is no known pathway that yield precisely a single extra centriole per template.

4. Fig 2H is a really important observation; could the authors include the frequency of this even in cell populations? This would strongly support the templated centriole overduplication model.

5. For single biological replicates, as in Fig 6C and D where 50 cells were scored for aneuploidy (expressed as % of cells), I do not know how the authors derived mean and standard deviation. This should be checked and corrected.

We would like to thank you very much for publishing our paper entitled “*GPER1 links estrogens to centrosome amplification and chromosomal instability in human colon cells*” in *Life Science Alliance*. We would like to thank all Reviewers again for their constructive feedback and positive attitude. We have carefully addressed all remaining comments of Reviewer #2 and #3 and revised the manuscript accordingly. In addition, we have implemented all points of the editor to meet the journal’s formatting guidelines. For text edits, we used the tracked change option (all edits of the main manuscript). Edits in the Material and Methods section and in the figure legends were labeled in yellow.

Concerning the comments of Reviewer #2:

1. We have followed the referee’s concerns. We toned down the outcome to the treatments in terms of centrosome numbers and mentioned the distinct differences seen using γ -tubulin and Cep135:

- p. 5: “The frequency distribution of centriole numbers based on counts with CP110 (Fig 2B)

was almost identical to that with γ -tubulin (Fig 2A). The distribution of Cep135 differed from those of γ -tubulin in HCT116 and even more in CCD 841 CoN cells with respect to individual treatments (Fig 2C).”

- p. 7: “Because (xeno)estrogens predominantly induced the formation of an odd number of centriole-positive centrosomes...”

2. We agree with the reviewer and rephrased the sentence according to the reviewer’s advice:

- p. 44 Figure legend to Figure S2: “**(B)** Graphical scheme for loss of centriole segregation at

M phase onset, which could indicate premature centriole disengagement...” [...] “Loss of centriole segregation during early mitosis would originate from ...”

- Material and Methods section on p. 28 “detection of centriole overduplication and premature disengagement / loss of centriole segregation”

3. The data presented for previous Figure 2A-D were indeed redundant for the γ -tubulin data presented in the tabular sheet of previous Figure S2A. We followed the reviewer’s advice and replaced the bar charts (previous Figure 2A-D) with the tabular data of previous Figure S2A-C (see current Figure 2A-C).

4. We replaced the term “evolvment” to “evolution” according to the reviewer’s advice.

Concerning the comments of Reviewer #3:

General comment: We agree with the reviewer and edit long and cumbersome sentences to make them more clear. Please find all changes in the tracked version.

1. We rephrased this sentence (in line with the other long sentences as mentioned above):
p. 7: “However, because of its superordinate role in regulating centrosome numbers, we cannot exclude that Plk4 overlays other (xeno)estrogen-triggered mechanisms.

2. We now describe the centriole disengagement / loss of centriole segregation (see Reviewer #2) experiment more clearly. See Material and Methods section “Detection of centriole overduplication and premature disengagement / loss of centriole segregation”. In addition, we have reprocessed the scheme in previous Figure S2E (now Fig S2B) according to the reviewer’s comments and to data from Wang et al. 2011 (REF 78 in the revised manuscript). Not least, we added representative images of metaphase cells for HCT116 and HCT-15 demonstrating Sas-6 signals that coincide with the PCM marker γ -tubulin (current Figure S2C and D). These images now accompany the graphs as suggested by the reviewer.

3. Concerning ‘three or five’ centrioles that may be generated in daughter cells (overduplication model, current Fig. 2E) I am unsure what the reviewer exactly mean. The reviewer is correct that there might be more than one centriole generated per template. The graphical scheme (current Figure 2E) is intended to reflect our experimental observations, i.e., that (xeno)estrogens mainly induce the generation of three centrosomes. However, the reviewer is absolute correct that there is no known pathway that yield precisely a single extra centriole per template. We rephrased the sentences as follows:

- p. 7: “At the end of mitosis, the parental centriole and its overduplicated procentrioles split apart (i.e. disengage), resulting in daughter cells with more than two centrioles. This pathway may lead to cells with amplified centrosomes after passage through the next cell cycle.”

- p. 8: “Our data rather support a ‘templating model’ for the extra centrosomes that arise in the presence of (xeno)estrogens. Whether this model explains the generation of predominantly three centrosomes needs to be investigated in future studies.”

4. The reviewer is absolutely right. The reviewer's comment made us aware that previous Figure 2H is misleading. It was originally intended to accompany the bar chart (Sas-6 fluorescence

intensities). However, the previous images showed a distinct phenotype, i.e., S-phase nuclei with two γ -tubulin signals and three Sas-6 signals (which would indeed support the ‘templating model’). However, we have not investigated (i.e. quantified) whether in fact only one additional centriole is formed per template (and not more). This exciting question will have to be investigated in future studies. We decide to exclude previous Figure 2H to avoid any misleading interpretations of the results. Representative images yielding to increased Sas-6 intensities at centrosomes (current Figure 2G) are shown in current Figure 2F. Of note, the proportion of the individual phenotypes shown in Figure 2F was not determined more precisely.

5. The reviewer is correct. We explained in the former point-by-point response letter to the referee #1 (point 19) that we were not able to provide p-values (because biological replicates are missing), but we tried to address the comments by providing standard deviations of the values (described in the previous Materials and Methods sections under “Statistics”). We therefore agree to correct the images (Fig. 5C, 6C, 6D, S5C, and S6F) back to the original version (i.e. without standard deviations). In addition, we deleted the term “mean” in the respective figure legends. This was indeed wrong and we thank the reviewer for his attentive look.

October 25, 2022

RE: Life Science Alliance Manuscript #LSA-2022-01499-TRR

Dr. Ailine Stolz
Federal Institute for Risk Assessment
Experimental Toxicology and ZEBET
Max-Dohrn-Str. 8-10
Berlin 10589
Germany

Dear Dr. Stolz,

Thank you for submitting your Research Article entitled "GPER1 links estrogens to centrosome amplification and chromosomal instability in human colon cells". It is a pleasure to let you know that your manuscript is now accepted for publication in Life Science Alliance. Congratulations on this interesting work.

DISTRIBUTION OF MATERIALS:

Again, congratulations on a very nice paper. I hope you found the review process to be constructive and are pleased with how the manuscript was handled editorially. We look forward to future exciting submissions from your lab.

Sincerely,
